# DeepfakeBench: A Comprehensive Benchmark of Deepfake Detection

**Zhiyuan Yan**[1], **Yong Zhang**[2], **Xinhang Yuan**[1], **Siwei Lyu**[3], **Baoyuan Wu**[1]*

[1]School of Data Science,
The Chinese University of Hong Kong, Shenzhen (CUHK-Shenzhen), China
[2]Tencent AI Lab
[3]Department of Computer Science and Engineering,
University at Buffalo, State University of New York, USA

## Abstract

A critical yet frequently overlooked challenge in the field of deepfake detection is the lack of a standardized, unified, comprehensive benchmark. This issue leads to unfair performance comparisons and potentially misleading results. Specifically, there is a lack of uniformity in data processing pipelines, resulting in inconsistent data inputs for detection models. Additionally, there are noticeable differences in experimental settings, and evaluation strategies and metrics lack standardization. To fill this gap, we present the first comprehensive benchmark for deepfake detection, called *DeepfakeBench*, which offers three key contributions: 1) a unified data management system to ensure consistent input across all detectors, 2) an integrated framework for state-of-the-art methods implementation, and 3) standardized evaluation metrics and protocols to promote transparency and reproducibility. Featuring an extensible, modular-based codebase, *DeepfakeBench* contains 15 state-of-the-art detection methods, 9 deepfake datasets, a series of deepfake detection evaluation protocols and analysis tools, as well as comprehensive evaluations. Moreover, we provide new insights based on extensive analysis of these evaluations from various perspectives (*e.g.*, data augmentations, backbones). We hope that our efforts could facilitate future research and foster innovation in this increasingly critical domain. All codes, evaluations, and analyses of our benchmark are publicly available at https://github.com/SCLBD/DeepfakeBench.

## 1 Introduction

Deepfake, widely recognized for its facial manipulation, has gained prominence as a technology capable of fabricating videos through the seamless superimposition of images. The surging popularity of deepfake technology in recent years can be attributed to its diverse applications, extending from entertainment and marketing to more complex usages. However, the proliferation of deepfake is not without risks. The same tools that enable creativity and innovation can be manipulated for malicious intent, undermining privacy, promoting misinformation, or eroding trust in digital media, *etc.*

Responding to the risks posed by deepfake contents, numerous deepfake detection methods [53, 22, 33, 32, 52, 3] have been developed to distinguish deepfake contents from real contents, which are generally categorized into three types: naive detector, spatial detector, and frequency detector. Despite rapid advancements in deepfake detection technologies, a significant challenge remains due to the lack of a standardized, unified, and comprehensive benchmark for a fair comparison among different detectors. This issue causes three major obstacles to the development of the deepfake detection field. **First**, there is a remarkable inconsistency in the training configurations and evaluation standards utilized in the field. This discrepancy inevitably leads to divergent outcomes, making a fair

---

*Corresponding author: Baoyuan Wu (wubaoyuan@cuhk.edu.cn)

37th Conference on Neural Information Processing Systems (NeurIPS 2023) Track on Datasets and Benchmarks.

| Model Type | Detectors | Backbone | Repositories | Reference |
|---|---|---|---|---|
| Naive Detector | MesoNet [1] | Designed CNN | https://github.com/DariusAf/MesoNet | WIFS-2018 |
| Naive Detector | MesoInception [1] | Designed CNN | https://github.com/DariusAf/MesoNet | WIFS-2018 |
| Naive Detector | CNN-Aug [48] | ResNet [16] | https://peterwang512.github.io/CNNDetection/ | CVPR-2020 |
| Naive Detector | EfficientNet-B4 [40] | EfficientNet [40] | https://github.com/lukemelas/EfficientNet-PyTorch | ICML-2019 |
| Naive Detector | Xception [33] | Xception [5] | https://github.com/ondyari/FaceForensics | ICCV-2019 |
| Spatial Detector | Capsule [29] | Designed Capsule [34] | https://github.com/nii-yamagishilab/Capsule-Forensics-v2 | ICASSP-2019 |
| Spatial Detector | DSP-FWA [22] | Xception [5] | https://github.com/danmohaha/CVPRW2019_Face_Artifacts | CVPRW-2019 |
| Spatial Detector | Face X-ray [20] | HRNet [46] | Unpublished code, reproduced by us | CVPR-2020 |
| Spatial Detector | FFD [6] | Xception [5] | cvlab.cse.msu.edu/project-ffd.html | CVPR-2020 |
| Spatial Detector | CORE [30] | Xception [5] | https://github.com/niyunsheng/CORE | CVPRW-2022 |
| Spatial Detector | RECCE [2] | Designed Networks | https://github.com/VISION-SJTU/RECCE | CVPR-2022 |
| Spatial Detector | UCF [50] | Xception [5] | Unpublished code, reproduced by us | ICCV-2023 |
| Frequency Detector | F3Net [32] | Xception [5] | Unpublished code, reproduced by us | ECCV-2020 |
| Frequency Detector | SPSL [26] | Xception [5] | Unpublished code, reproduced by us | CVPR-2021 |
| Frequency Detector | SRM [27] | Xception [5] | Unpublished code, reproduced by us | CVPR-2021 |

Table 1: *Summary of the compared deepfake detectors. For detectors without publicly available repositories, we undertake careful re-implementation, adhering to the instructions specified in the original papers.*

comparison difficult. **Second**, the source codes of many methods are not publicly released, which could be detrimental to the reproducibility and comparability of their reported results. **Third**, we find that the detection performance can be significantly influenced by several seemingly inconspicuous factors, *e.g.*, the number of selected frames in a video. Since the settings of these factors are not uniform and their impacts are not thoroughly studied in most existing works, the reported results and corresponding claims may be biased or misleading.

To bridge this gap, we build the first comprehensive benchmark, called **DeepfakeBench**, offering a unified platform for deepfake detection. Our main contributions are threefold. **1) An extensible modular-based codebase:** Our codebase consists of three main modules. The data processing module provides a unified data management module to guarantee consistency across all detection inputs, such that alleviating the time-consuming data processing and evaluation. The training module provides a modular framework to implement state-of-the-art detection algorithms, facilitating direct comparisons among different detection algorithms. The evaluation and analysis module provides several widely adopted evaluation metrics and rich analysis tools to facilitate further evaluations and analysis. **2) Comprehensive evaluations:** We evaluate 15 state-of-the-art detectors with 9 deepfake datasets under a wide range of evaluation settings, providing a holistic performance evaluation of each detector. Moreover, we establish a unified evaluation protocol that enhances the transparency and reproducibility of performance evaluation. **3) Extensive analysis and new insights:** We provide extensive analysis from various perspectives, not only analyzing the effects of existing algorithms but also uncovering new insights to inspire new technologies. **In summary**, we believe *DeepfakeBench* could constitute a substantial step towards calibrating the current progress in the deepfake detection field and promoting more innovative explorations in the future.

## 2   Related Work

**Deepfake Generation**    Deepfake technology, which generally centers on the artificial manipulation of facial imagery, has made considerable strides from its rudimentary roots. Starting in 2017, learning-based manipulation techniques have made significant advancements, with two prominent methods gaining considerable attention: Face-Swapping and Face-Reenactment. **1) Face-swapping** constitutes a significant category of deepfake generation. These techniques typically involve autoencoder-based manipulations, which are based on two autoencoders with a shared encoder and two different decoders. The autoencoder output is then blended with the rest of the image to create the forgery image. Notable face-swapping datasets of this approach include UADFV [21], FF-DF [7], CelebDF [23], DFD [9], DFDC [8], DeeperForensics-1.0 [17], and ForgeryNet [31]. **2) Face-reenactment** is characterized by graphics-based manipulation techniques that modify source faces imitating the expressions of a different face. NeuralTextures [41] and Face2Face [42], utilized in FaceForensics++, stand out as standard face-reenactment methods. Face2Face uses key facial points to generate varied expressions, while NeuralTexture uses rendered images from a 3D face model to migrate expressions.

**Deepfake Detection**    Current deepfake detection can be broadly divided into three categories: naive detector, spatial detector, and frequency detector. **1) Naive detector** employs CNNs to directly distinguish deepfake content from authentic data. Numerous CNN-based binary classifiers have been proposed, *e.g.*, MesoNet [1] and Xception [33]. **2) Spatial detector** delves deeper into specific representation such as forgery region location [28], capsule network [29], disentanglement learning [50, 24], image reconstruction [2], erasing technology [45], *etc*. Besides, some other

| Feature / Paper | DeepfakeBench | Paper [25] |
|---|---|---|
| Scope of Deepfake | Face-swapping + Diffusion + GAN | Face-swapping |
| Number of Detectors | 15 | 11 |
| Number of Datasets | 9 | 8 |
| Code Open Source | ✓ | Not yet |
| Modular and Extensible Codebase | ✓ | - |
| User-Friendly APIs | ✓ | - |
| Customizable Preprocessing Module | ✓ | - |
| Unified Training Framework | ✓ | - |
| Rich Analysis Tools | ✓ | ✓ |
| Analysis of FLOPs | - | ✓ |
| Evaluation Metrics | AUC, AP, ACC, EER, Precision, Recall | AUC |

Table 2: *Comprehensive comparison of our benchmark with existing benchmark [25].*

methods specifically focus on the detection of blending artifacts [22, 20, 3], generating forged images during training in a self-supervised manner to boost detector generalization. **3) Frequency detector** addresses this limitation by focusing on the frequency domain for forgery detection [13, 32, 26, 27]. SPSL [26] and SRM [27] are other examples of frequency detectors that utilize phase spectrum analysis and high-frequency noises, respectively. Qian *et al.* [32] propose the use of learnable filters for adaptive mining of frequency forgery clues using frequency-aware image decomposition.

**Related Deepfake Surveys and Benchmarks**    The growing implications of deepfake technology have sparked extensive research, resulting in the establishment of several surveys and dataset benchmarks in the field. **1) Surveys** provide a detailed examination of various facets of deepfake technology. For instance, Westerlund *et al.* [49] present a thorough analysis of deepfake, emphasizing its legal and ethical dimensions. Tolosana *et al.* [43] furnish a comprehensive review of face manipulation techniques, including deepfake methods, along with approaches to detect such manipulations. **2) Benchmarks** in this field have emerged as essential tools to provide realistic forgery datasets. For instance, FaceForensics++ (FF++) [33] serves as a prominent benchmark, offering high-quality manipulated videos and a variety of forgery types. The Deepfake Detection Challenge Dataset (DFDC) [10] introduces a diverse range of actors across different scenarios.

While these benchmarking methodologies have made significant contributions, they specifically focus on their own datasets, without offering a standardized way to handle data across different datasets, which may lead to inconsistencies and obstacles to fair comparisons. Also, the lack of a unified framework in some benchmarks could lead to variations in training strategies, settings, and augmentations, which may result in discrepancies in the outcomes. Furthermore, the provision of comprehensive analytical tools is not always prominent, which might restrict the depth of analysis on the potential impacts of different factors. One notable work [25] aims to build a benchmark for evaluating various detectors under different datasets. Another recent work [18] introduces a benchmark centered around detecting GAN-generated images using continual learning. However, these two benchmarks still lack a modular, extensible, and comprehensive codebase that includes data preprocessing, unified settings, training modules, evaluations, and a series of analytical tools. *DeepfakeBench*, on the other hand, presents a concise but comprehensive benchmark. Its contributions are threefold: introducing a unified data management system for consistency, offering an integrated framework for implementing advanced methods, and analyzing the related factors with a series of analysis tools. Detailed comparisons between our *DeepfakeBench* and [25] are shown in Tab.2.

## 3   Our Benchmark

### 3.1   Datasets and Detectors

***Datasets***    Our benchmark currently incorporates a collection of 9 widely recognized and extensively used datasets in the realm of deepfake detection: FaceForensics++ (FF++) [33], CelebDF-v1 (CDFv1) [23], CelebDF-v2 (CDFv2) [23], DeepFakeDetection (DFD) [9], DeepFake Detection Challenge Preview (DFDC-P) [11], DeepFake Detection Challenge (DFDC) [10], UADFV [21], FaceShifter (Fsh) [19], and DeeperForensics-1.0 (DF-1.0) [17]. Notably, FF++ contains 4 types of manipulation methods: Deepfakes (FF-DF) [7], Face2Face (FF-F2F) [42], FaceSwap (FF-FS) [14], NeuralTextures (FF-NT) [41]. There are three versions of FF++ in terms of compression level, *i.e.*,

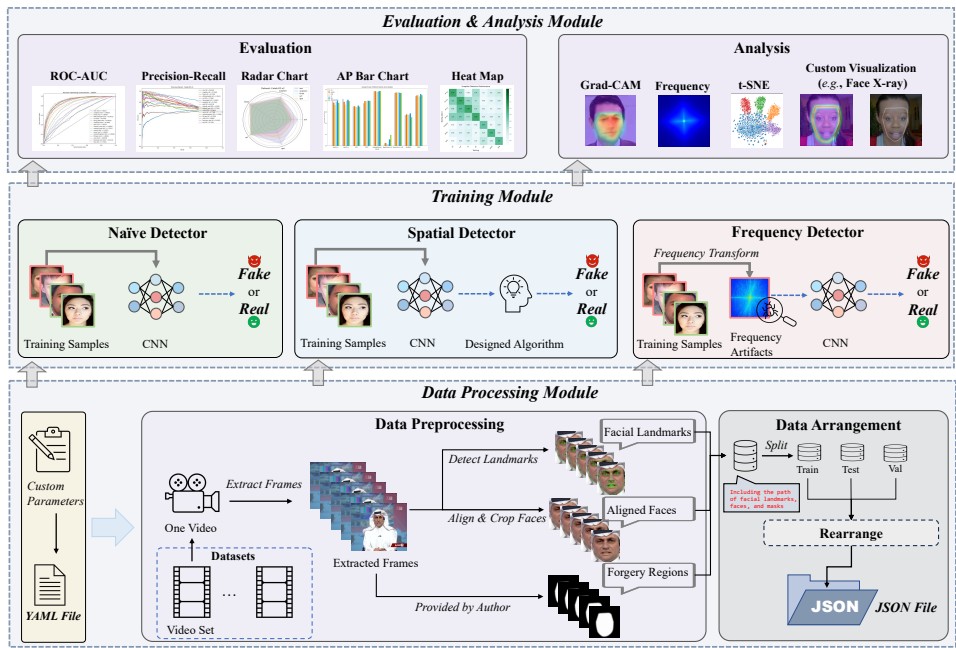

Figure 1: The general structure of the modular-based codebase of *DeepfakeBench*.

raw, lightly compressed (c23), and heavily compressed (c40). The detailed descriptions of each dataset are presented in the Sec. A.3 of the **Appendix**. Typically, FF++ is employed for model training, while the rest are frequently used as testing data. However, our benchmark allows users to select their combinations of training and testing data, thus encouraging custom experimentation.

It is notable that, although these datasets have been widely used in the community, they are not usually provided in a readily accessible and combined format. It often requires a substantial investment of time and effort in data sourcing, pre-processing (*e.g.*, frame extraction, face cropping, and face alignment), and organization of the raw datasets, which are often organized in diverse structures. This considerable data preparation overhead often diverts researchers' attention away from the core tasks like methodology design and experimental evaluations. To tackle this challenge, our benchmark offers a collection of well-processed and systematically organized datasets, allowing researchers to devote more time to the core tasks. Additionally, our benchmark enriches some datasets (*e.g.*, FF++ [33] and DFD [9]), by including mask data (*i.e.*, the forgery region) that is aligned with the respective facial images in these datasets. It could facilitate more comprehensive deepfake detection studies. **In summary**, our benchmark provides a unified, user-friendly, and diversified data resource for the deepfake detection community. It eliminates the cumbersome task of data preparation and allows researchers to concentrate more on innovating effective deepfake detection methods.

***Detectors*** Our benchmark has implemented a total of 15 established deepfake detection algorithms, as detailed in Tab. 1. The selection of these algorithms is guided by three criteria. **First**, we prioritize methods that hold a classic status (*e.g.*, Xception), or those considered advanced, typically published in recent top-tier conferences or journals in computer vision or machine learning. **Second**, our benchmark classifies detectors into three categories: naive detectors, spatial detectors, and frequency detectors. Our primary emphasis is on image forgery detection, hence, temporal-based detectors have not yet been incorporated. Moreover, we have refrained from including traditional detectors (*e.g.*, Headpose [51]) due to their limited scalability to large-scale datasets, making them less suitable for our benchmark's objectives. **Third**, we aim to include methods that are straightforward to implement and reproduce. We notice that several existing methods involve a series of steps, some of which are reliant on third-party algorithms or heuristic strategies. These methods usually have numerous hyper-parameters and are fraught with uncertainty, making their implementation and reproduction challenging. Therefore, these methods without open-source codes are intentionally excluded from our benchmark. However, it is important to note that there are also some non-open-source methods we employed that are derived from the code directly provided by their respective authors.

## 3.2 Codebase

We have built an extensible modular-based codebase as the basis of *DeepfakeBench*. As shown in Fig. 1, it consists of three core modules, including *Data Processing Module*, *Training Module*, and *Evaluation and Analysis Module*.

***Data Processing Module***    The *Data Processing Module* includes two pivotal sub-modules that automate the data processing sequence, namely the *Data Preprocessing* and *Data Arrangement* sub-modules.    **1) Data preprocessing** sub-module presents a streamlined solution. First, Users are provided with a *YAML* configuration file, enabling them to tailor the preprocessing steps to their specific requirements. Second, we furnish a unified preprocessing script, which includes frame extraction, face cropping, face alignment, mask cropping, and landmark generation. **2) Data arrangement** sub-module further augments the convenience of data management. This sub-module comprises a suite of *JSON* files for each dataset. Users can execute a rearranged script to create a unified *JSON* file for each dataset. This unified file provides access to the corresponding training, testing, and validation sets, along with other information such as the frames, landmarks, masks, *etc*, related to each dataset.

***Training Module***    The *Training Module* currently accommodates 15 detectors across three categories: naive detector, spatial detector, and frequency detector, all of which are shown in Tab. 1. **1) Naive detector** leverages various CNN architectures to directly detect forgeries without relying on additional manually designed features. **2) Spatial detector** builds upon the backbone of CNNs used in the Naive Detector and further explores manual-designed algorithms to detect deepfake. **3) Frequency detector** focuses on utilizing information from the frequency domain and extracting frequency artifacts for detection. Each detector implemented in our benchmark is managed in a streamlined and efficient way, with a *YAML* config file created for each one. This allows users to easily set their desired parameters, *e.g.*, batch size, learning rate, *etc.* These detectors are trained on a unified trainer that records the metrics and losses during the training and evaluation process. Thus, the training and evaluation processes, logging, and visualization are handled automatically, eliminating the need for manual specification.

***Evaluation and Analysis Module***    **For evaluation**, we employ 4 widely used evaluation metrics: accuracy (ACC), the area under the ROC curve (AUC), average precision (AP), and equal error rate (EER) Besides, it is notable that there is an inconsistency in the usage of these evaluation metrics in the community, some are at the frame level, while others are at the video level, leading to unfair comparisons. Our benchmark currently adopts the frame level evaluation to build a fair basis for comparison among detectors. In addition to the evaluation values of these metrics, we also provide several visualizations to facilitate performance comparisons, *e.g.*, the ROC-AUC curve, radar chart, and histogram. **For analysis**, we provide various visualization tools to gain deeper insights into the detectors' performance. For example, Grad-CAM [36] is used to highlight the potential forgery regions detected by the models, providing interpretability and assisting in understanding the underlying reasoning for the model's predictions. To explore the learned features and representations, we employ t-SNE visualization [44]. Furthermore, we offer custom visualizations tailored to specific detectors. For example, for Face X-ray [20], we provide visualizations of the detection boundary of the face, as described in its original paper (see the top-right corner of Fig. 1).

## 4 Evaluations and Analysis

### 4.1 Experimental Setup

In the data processing, face detection, face cropping, and alignment are performed using DLIB [35]. The aligned faces are resized to $256 \times 256$ for both the training and testing. In the training module, we employ the Adam optimization algorithm with a learning rate of 0.0002. The batch size is fixed at 32 for all experiments. We sample 32 frames for each video for training and testing. We primarily leverage pre-trained backbones from ImageNet if feasible. Otherwise, we resort to initializing the remaining weights using a normal distribution. We also apply widely used data augmentations, *i.e.,* image compression, horizontal flip, rotation, Gaussian blur, and random brightness contrast. In terms of evaluation, we compute the average value of the top-3 metrics (*e.g.*, average top-3 AUC) as our evaluation metric. We also report other metrics (*i.e.*, AP, EER, Precision, and Recall) in the Sec. A.3 of the **Appendix**. Further details of dataset configuration, algorithms implementation, and full training details can be seen in the Sec. A.1, Sec. A.2, and Sec. A.3 of the **Appendix**, respectively.

| Type | Detector | Backbone | Within Domain Evaluation | | | | | | | | Cross Domain Evaluation | | | | | | | | | |
|---|---|---|---|---|---|---|---|---|---|---|---|---|---|---|---|---|---|---|---|---|
| | | | FF-c23 | FF-c40 | FF-DF | FF-F2F | FF-FS | FF-NT | Avg. | Top3 | CDFv1 | CDFv2 | DF-1.0 | DFD | DFDC | DFDCP | Fsh | UADFV | Avg. | Top3 |
| Naive | Meso4 [1] | MesoNet | 0.6077 | 0.5920 | 0.6771 | 0.6170 | 0.5946 | 0.5701 | 0.6097 | 0 | 0.7358 | 0.6091 | 0.9113 | 0.5481 | 0.5560 | 0.5994 | 0.5660 | 0.7150 | 0.6551 | 1 |
| Naive | MesoIncep [1] | MesoNet | 0.7583 | 0.7278 | 0.8542 | 0.8087 | 0.7421 | 0.6517 | 0.7571 | 0 | 0.7366 | 0.6966 | 0.9233 | 0.6069 | 0.6226 | 0.7561 | 0.6438 | 0.9049 | 0.7364 | 3 |
| Naive | CNN-Aug [48] | ResNet | 0.8493 | 0.7846 | 0.9048 | 0.8788 | 0.9026 | 0.7313 | 0.8419 | 0 | 0.7420 | 0.7027 | 0.7993 | 0.6464 | 0.6361 | 0.6170 | 0.5985 | 0.8739 | 0.7020 | 0 |
| Naive | Xception [33] | Xception | 0.9637 | 0.8261 | 0.9799 | 0.9785 | 0.9833 | 0.9385 | 0.9450 | 4 | 0.7794 | 0.7365 | 0.8341 | **0.8163** | 0.7077 | 0.7374 | 0.6249 | 0.9379 | 0.7718 | 2 |
| Naive | EfficientB4 [40] | Efficient | 0.9567 | 0.8150 | 0.9757 | 0.9758 | 0.9797 | 0.9308 | 0.9389 | 3 | 0.7909 | 0.7487 | 0.8330 | 0.8148 | 0.6955 | 0.7283 | 0.6162 | 0.9472 | 0.7718 | 3 |
| Spatial | Capsule [29] | Capsule | 0.8421 | 0.7040 | 0.8669 | 0.8634 | 0.8734 | 0.7804 | 0.8217 | 0 | 0.7909 | 0.7472 | 0.9107 | 0.6841 | 0.6465 | 0.6568 | 0.6465 | 0.9078 | 0.7488 | 2 |
| Spatial | FWA [22] | Xception | 0.8765 | 0.7357 | 0.9210 | 0.9000 | 0.8843 | 0.8120 | 0.8549 | 0 | 0.7897 | 0.6680 | **0.9334** | 0.7403 | 0.6132 | 0.6375 | 0.5551 | 0.8539 | 0.7239 | 1 |
| Spatial | X-ray [20] | HRNet | 0.9592 | 0.7925 | 0.9794 | **0.9872** | 0.9871 | 0.9290 | 0.9391 | 3 | 0.7093 | 0.6786 | 0.5531 | 0.7655 | 0.6326 | 0.6942 | **0.6553** | 0.8989 | 0.6985 | 0 |
| Spatial | FFD [6] | Xception | 0.9624 | 0.8237 | 0.9803 | 0.9784 | 0.9853 | 0.9306 | 0.9434 | 1 | 0.7840 | 0.7435 | 0.8609 | 0.8024 | 0.7029 | 0.7426 | 0.6056 | 0.9450 | 0.7733 | 1 |
| Spatial | CORE [30] | Xception | 0.9638 | 0.8194 | 0.9787 | 0.9803 | 0.9823 | 0.9339 | 0.9431 | 2 | 0.7798 | 0.7428 | 0.8475 | 0.8018 | 0.7049 | 0.7341 | 0.6032 | 0.9412 | 0.7694 | 0 |
| Spatial | Recce [2] | Designed | 0.9621 | 0.8190 | 0.9797 | 0.9779 | 0.9785 | 0.9357 | 0.9422 | 1 | 0.7677 | 0.7319 | 0.7985 | 0.8119 | 0.7133 | 0.7419 | 0.6095 | 0.9446 | 0.7649 | 2 |
| Spatial | UCF [50] | Xception | **0.9705** | **0.8399** | **0.9883** | 0.9840 | **0.9896** | **0.9441** | **0.9527** | **6** | 0.7793 | 0.7527 | 0.8241 | 0.8074 | **0.7191** | **0.7594** | 0.6462 | **0.9528** | 0.7801 | **5** |
| Frequency | F3Net [32] | Xception | 0.9635 | 0.8271 | 0.9793 | 0.9796 | 0.9844 | 0.9354 | 0.9449 | 1 | 0.7769 | 0.7352 | 0.8431 | 0.7975 | 0.7021 | 0.7354 | 0.5914 | 0.9347 | 0.7645 | 0 |
| Frequency | SPSL [26] | Xception | 0.9610 | 0.8174 | 0.9781 | 0.9754 | 0.9829 | 0.9299 | 0.9408 | 0 | **0.8150** | **0.7650** | 0.8767 | 0.8122 | 0.7040 | 0.7408 | 0.6437 | 0.9424 | **0.7875** | 3 |
| Frequency | SRM [27] | Xception | 0.9576 | 0.8114 | 0.9733 | 0.9696 | 0.9740 | 0.9295 | 0.9359 | 0 | 0.7926 | 0.7552 | 0.8638 | 0.8120 | 0.6995 | 0.7408 | 0.6014 | 0.9427 | 0.7760 | 2 |

Table 3: *Within-domain and cross-domain evaluations using the AUC metric. All detectors are trained on FF-c23 and evaluated on other data. "Avg." donates the average AUC for within-domain and cross-domain evaluation, and the overall results. "Top3" represents the count of each method ranks within the top-3 across all testing datasets. The best-performing method for each column is highlighted in* red*.*

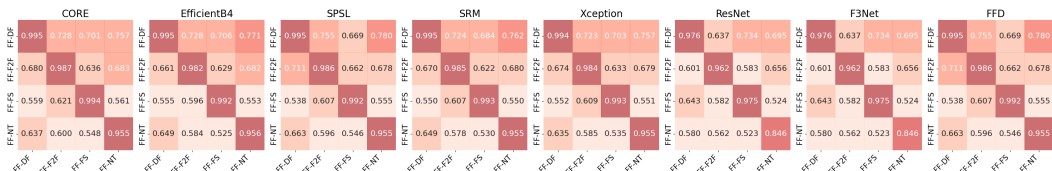

Figure 2: *Visualization of heat maps showing the cross-manipulation evaluation results. The color represents the AUC performance index of the corresponding detector under specific test data, and the darker the color, the better the performance. All heat maps use a uniform color scale for performance comparison.*

## 4.2 Evaluations

In this section, we focus on performing two types of evaluations: **1) within-domain and cross-domain evaluation**, and **2) cross-manipulation evaluation**. The purpose of the within-domain evaluation is to assess the performance of the model within the same dataset, while cross-domain evaluation involves testing the model on different datasets. We also perform cross-manipulation evaluation to evaluate the model's performance on different forgeries under the same dataset.

***Within-Domain and Cross-Domain Evaluations*** In this evaluation, we specifically train the model using FF++ (c23) as the default training dataset. Subsequently, we evaluate the model on a total of 14 different testing datasets, with 6 datasets for within-domain evaluation and 8 datasets for cross-domain evaluation. Tab. 3 provides an extensive evaluation of various detectors, divided into Naive, Spatial, and Frequency types, based on both within-domain and cross-domain tests. Regarding the results in Tab. 3, we observe that, for the within-domain evaluations, a majority of the detectors performed commendably, evidenced by high within-domain AUC. Remarkably, detectors such as UCF, Xception, EfficientB4, and F3Net registered significant average scores, specifically 95.37%, 94.50%, 93.89%, and 94.49% respectively. Furthermore, an unexpected revelation comes from the performance of Naive Detectors. Astonishingly, Naive Detectors (*e.g.*, Xception and EfficientB4), which essentially rely on a straightforward CNN classifier, register high AUC values that are comparable to more sophisticated algorithms. This could potentially suggest that the performance leap from advanced state-of-the-art methods to Naive Detectors might not be as substantial as perceived, particularly in consistent settings (*e.g.*, pre-training or data augmentation). In other words, the performance gap could be a product of these additional factors rather than the intrinsic superiority of the method. To delve deeper into this phenomenon, we will investigate the impact of data augmentation, backbone architecture, pre-training, and the number of training frames in the following section (see Sec. 4.3).

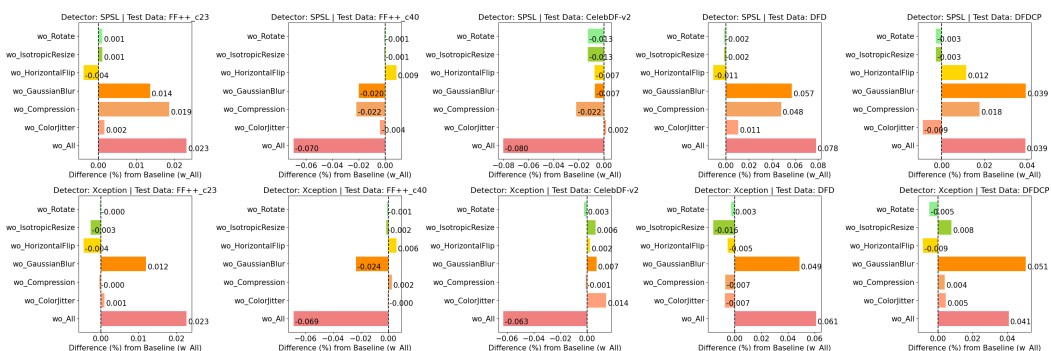

Figure 3: *Visualization of different augmentation methods. We apply two detectors, one in the spatial domain (Xception) and one in the frequency domain (SPSL), and then use 8 different augmentation strategies to measure the effect on 5 test datasets.*

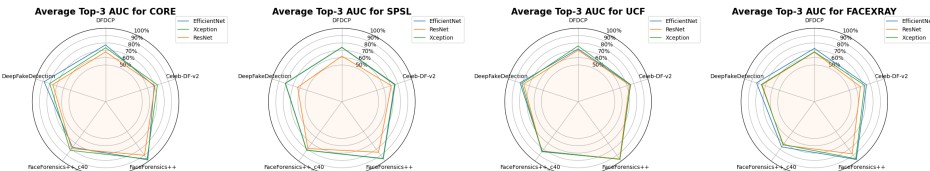

Figure 4: *Visualization of the performance of 3 different backbones, ResNet, EfficientNet-B4, and Xception, across 4 different detectors, CORE, SPSL, UCF, and Face X-ray. The evaluation is conducted using the AUC metric, following the settings described in the previous section.*

***Cross-Manipulation Evaluations*** We also conduct a cross-manipulation evaluation to assess the model's performance on various manipulation forgeries within the same dataset (FF++ [33]). In this evaluation, only the forgery algorithm is altered. Other factors such as background and identity remain consistent across all the different forgeries. Fig. 2 compares the cross-manipulation detection performance of 10 detectors. Upon examining the figure, it becomes evident that the issue of generalization is prominent. While detectors such as CORE, EfficientB4, SPSL, SRM, and Xception exhibit excellent performance on the FF-DF test data when trained on FF-DF, their performance significantly deteriorates when faced with FF-FS forgeries. Furthermore, the "FT-NT" test data poses challenges for almost all detectors, as reflected by the diminished AUC values in this category throughout the heatmaps. In contrast, the "FT-DF" test data emerged as a comparatively facile challenge for the detectors. **In summary**, the varying nature of forgeries highlights a significant generalization gap. Models trained on specific forgeries often struggle to adapt to other unseen forgeries. This underscores the importance of training models to recognize generic forgery artifacts to better combat unseen forgery types.

### 4.3 Analysis

**Effect of Data Augmentation** We assess the influence of various augmentation techniques on the performance of forgery detectors in this section. Specifically, we investigate the impact of rotations, horizontal flips, image compression, isotropic scaling, color jitter, and Gaussian blur on two prototypical detectors: one from the spatial domain (Xception) and one from the frequency domain (SPSL). Fig. 3 compares the performance when training these detectors with all data augmentations (denoted as "w_All"), without any data augmentations ("wo_All"), and without a specific augmentation.

Our findings can be summarized into three main observations: **First**, in the case of within-domain evaluation (as seen in the FF++_c23 dataset), removing all augmentations appears to improve detector performance by approximately 2% for both Xception and SPSL, suggesting that most augmentations may have a negative impact within this context. **Second**, for evaluations involving compressed data (FF++_c40), certain augmentations such as Gaussian blur demonstrate effectiveness in both Xception and SPSL detectors, as they simulate the effects of compression on the data during training. **Third**, in the context of cross-domain evaluations (CelebDF-v2, DFD, and DFDCP), operations like compression and blur may significantly degrade the performance of SPSL in the DFD and DFDCP datasets, possibly due to their tendency to obscure high-frequency details. Similar negative effects of

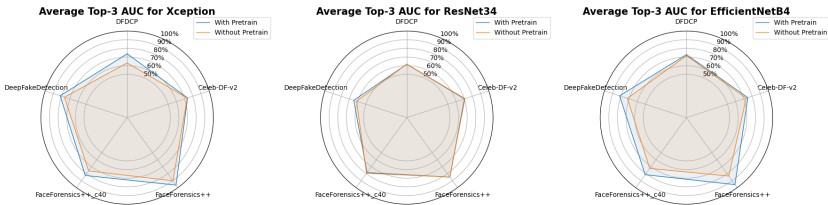

Figure 5: *Visualization of the effect of pre-trained weights on three different architectures. The evaluation is conducted using the AUC metric, following the settings described in the previous section.*

the blur operation are observed for Xception, likely as it diminishes the visibility of visual artifacts. These findings underscore the need for further exploration into identifying a universally beneficial augmentation that can be effectively utilized across a wide range of detectors in generalization scenarios, irrespective of their specific attributes or datasets.

| Model | FF++_c23 | FF++_c40 | CDF-v2 | DFD | DFDCP | UADFV | Average |
|---|---|---|---|---|---|---|---|
| ResNet | 0.8493 | 0.7846 | 0.7027 | 0.6464 | 0.6170 | 0.8739 | 0.7456 |
| ResNet-DSC | 0.8968 | 0.8048 | 0.7582 | 0.7006 | 0.6766 | 0.8895 | 0.7877 |
| Improvement (%) | +5.60% | +2.57% | +7.90% | +8.39% | +9.64% | +1.78% | +5.64% |

Table 4: *Ablation study regarding the effectiveness of the depthwise separable convolution module (DSC) for ResNet. The models are trained on FF++_c23 and tested on other datasets. The metric is the frame-level AUC.*

**Effect of Backbone Architecture**     We here investigate the impact of different backbone architectures on the performance of forgery detection models. Specifically, we compare the performance of three popular backbones: Xception, EfficientNet-B4, and ResNet34. Each backbone is integrated into the detection model, and its performance is evaluated on both within-domain and cross-domain datasets (see Fig. 4). Our findings reveal that Xception and EfficientNet-B4 consistently outperform ResNet34, despite having a similar number of parameters. This indicates that the choice of backbone architecture plays a crucial role in detector performance, especially when evaluating the DeepfakeDetection dataset using CORE. **In summary**, these results highlight the critical role of carefully selecting a suitable backbone architecture in the design of deepfake detection models. Further research in this direction holds the potential for advancing the field in the future.

**Additional In-depth Analysis towards the Effect of Backbone Architecture**     When analyzing the effect of backbone architecture, our analysis in Sec. 4.3 shows that Xception and EfficientNet-B4 work better than ResNet-34. Given the three architectures have similar numbers of parameters, we are curious about why there exists an obvious performance gap among the three architectures. Here, we dive deeper to explore the possible reasons.

After our preliminary investigation, we found that the reasons are related to two factors, namely architecture and models' scale. **First**, we identify a common module in EfficientNet and Xception that is not present in ResNet, namely the **depthwise separable convolution module**. We hypothesize that this module might be contributing to the performance advantage. To evaluate this, we insert this module into ResNet, replacing only the first convolutional layer. Experiments demonstrate significant improvements on many test datasets (as shown in Tab. 4). **Second**, upon closer scrutiny, additional factors that might exert an impact on the ultimate performance come to light. These encompass the number of layers within the model architecture as well as the number of parameters associated with it. Referring to Tab. 8 in the **Appendix**, it becomes evident that the parameter numbers remain comparable among the three models. Subsequently, a comprehensive exploration is conducted to assess the impact of layer numbers. This assessment involves a diverse range of ResNet variants, including ResNet 50 and ResNet 152. Results in Tab. 9 in our **Appendix** uncover that ResNet 50, characterized by a greater number of layers in comparison to ResNet 34, yields a substantial enhancement in performance. However, when confronted with a higher layer count, as exemplified by ResNet 152, the extent of improvement becomes restricted.

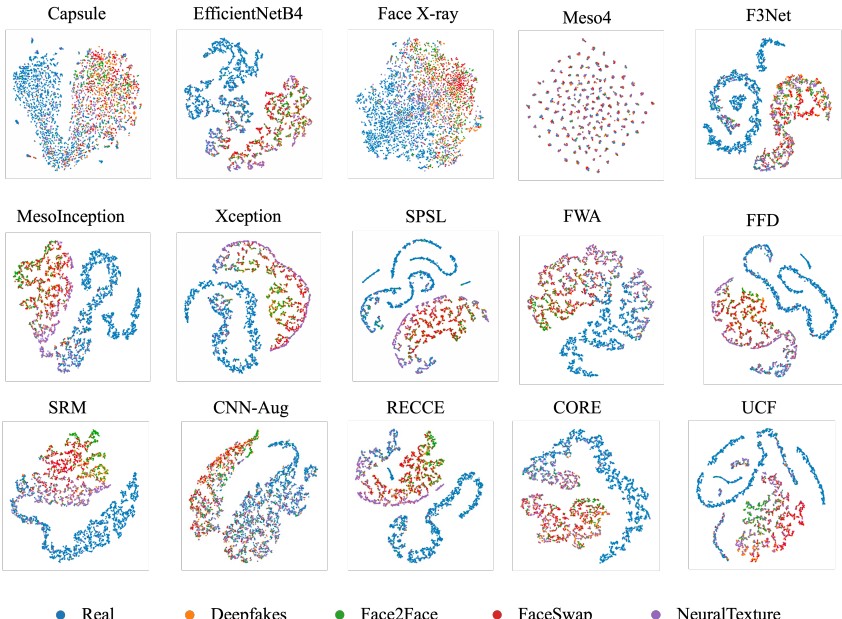

Figure 6: t-SNE visualization for each detector. These detectors are trained and tested on FF++ (c23).

**Effect of Pre-training of the Backbone** This analysis focuses on the impact of pre-training on forgery detection models. Following the previous section, we analyze three typical architectures: Xception, EfficientNetB4, and ResNet34. Fig. 5 reveals that the pre-trained models can largely outperform their non-pre-trained counterparts, especially in the case of Xception (about 10% in DFDCP) and EfficientB4 (about 10% in DeepFakeDetection). This can be attributed to the ability of pre-trained models to capture and leverage meaningful low-level features. However, the benefits of pre-training are less pronounced for ResNet34, mainly due to its architectural design, which may not fully exploit the advantages offered by pre-trained weights. **Overall**, our findings underscore the importance of both architectural choices and the utilization of pre-trained weights in achieving optimal forgery detection performance.

**Visualizing Representations** Deepfake detection can be considered a representation learning problem, where detectors learn representations through their backbones and employ various classification algorithms. It is crucial to assess whether the learned representations align with the expectations. To accomplish this, we utilize t-SNE [44] for analysis, which allows us to visualize the representation.

We examine t-SNE visualization from two perspectives. First, we assess whether the detectors can accurately differentiate between real and fake samples. This is achieved by assigning labels to the points in the t-SNE plot based on their corresponding ground truth. Second, we delve deeper into the fake category and investigate whether the models capture common features across different forgery types rather than being overfitted to specific forgeries. To conduct this analysis, we train and test each detector on the FF++ (c23) dataset and visualize the t-SNE representation using the test data. Also, we visualize all the samples with their corresponding labels, where the Deepfakes, Face2Face, FaceSwap, and NeuralTextures represent different forgery types in FF++. For visualization purposes, we randomly select 5000 samples, with an equal distribution of 2500 real and 2500 fake samples. Default parameters are used for t-SNE.

From the t-SNE results shown in Fig. 6, we observe that different detectors learn distinct feature representations in the visualized space. Notably, the results indicate that Meso4 struggles to differentiate between real and fake samples, as the two categories overlap and cannot be clearly distinguished.

## 5 Conclusions, Future Plans, and Societal Impacts

**Conclusions** We have developed *DeepfakeBench*, a groundbreaking and comprehensive framework, emphasizing the benefits of a modular architecture, including extensibility, maintainability, fairness,

and analytical capability. We hope that *DeepfakeBench* could contribute to the deepfake detection community in various ways. **First**, it provides a concise yet comprehensive platform that incorporates a tailored data processing pipeline, and accommodates a wide range of detectors, while also facilitating a fair and standardized comparison among various models. **Second**, it assists researchers in swiftly comparing their new methods with existing ones, thereby facilitating faster development and iterations. **Last**, the in-depth analysis and comprehensive evaluations performed through our benchmark have the potential to inspire novel research problems and drive future advancements in the field.

**Limitations and Future Plans** To date, *DeepfakeBench* primarily focuses on providing algorithms and evaluations at the frame level. We will further enhance the benchmark by incorporating video-level detectors and evaluation metrics. This expansion will enable a more comprehensive assessment of forgery detection performance, considering the temporal dynamics and context within videos. Besides, we also plan to carry out more evaluations for detecting images directly produced by diffusion or GANs, using the existing benchmark. In the current version, we have provided the visualizations and analysis for GAN-generated and diffusion-generated data in the frequency domain (see Sec. A.4 in the Appendix). Furthermore, we aim to include a wider range of typical detectors and datasets to offer a more comprehensive platform for evaluating the performance of detectors. *DeepfakeBench* will continue to evolve as a valuable resource for researchers, facilitating the development of advanced deepfake detection technologies.

**Societal Impact and Ethical Issue** The potential ethical issue lies in the risk that malicious actors might exploit *DeepfakeBench* to refine deepfakes to evade detection. **1) Inherent challenge with benchmarking:** *DeepfakeBench*, like any benchmark created for positive intent, could inadvertently provide a blueprint for these actors due to its transparent nature. **2) Potential solutions and forward path:** As solutions, we are contemplating controlled access for users and are committed to the dynamic evolution of DeepfakeBench to ensure it remains robust against emerging threats.

## 6   Contents in Appendix

The Appendix accompanying this paper provides additional details. The Appendix is organized as follows: **1) Details of data processing** This section provides further elaboration on the data processing steps, including face detection, face cropping, alignment, and *etc*. **2) Details of algorithms implementation and visualizations** This section dives into the implementation details of the algorithms used in the study. It also includes additional visualizations to help readers gain a deeper understanding of the experimental results. **3) Training details and full experimental results**: This section presents comprehensive details of the training process, including additional evaluation metrics beyond those reported in the main paper. **4) Other analysis results**: This section conducts analysis on some parts that are not analyzed in detail in the main text, such as analyzing and visualizing the frequency domain analysis of images generated by GAN and diffusion, etc.

## 7   Acknowledgement

This work is supported by the National Natural Science Foundation of China under grant No. 62076213, Shenzhen Science and Technology Program under grant No. RCYX20210609103057050, No. ZDSYS20211021111415025, No. GXWD20201231105722002-20200901175001001, and the Guangdong Provincial Key Laboratory of Big Data Computing, the Chinese University of Hong Kong, Shenzhen.

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

# A  Appendix

## A.1  Details of Data Processing

This section introduces a data preprocessing script tailored for deepfake datasets. This script incorporates a series of fundamental steps, including **face detection**, **face cropping**, **face alignment**, and various other preprocessing operations. These steps are of utmost importance as they facilitate the acquisition of consistent face images, thereby ensuring the effectiveness and reliability of subsequent analysis and model training. The following subsections describe each step in detail.

**Overall Workflow**    The preprocessing script follows a sequential workflow. It starts by detecting faces in each video frame using the Dlib [35] face detection algorithm. Once the faces are detected, the script proceeds to **align and crop the faces** based on the detected facial landmarks. If a mask video file is provided, the script also extracts and saves the masks for each aligned face. **The face images, landmarks, and masks are saved in separate folders but in the same directory for further analysis.** The preprocessing script also supports **parallel processing**, which enables multiple videos to be processed simultaneously, improving the overall processing speed. Each video is processed independently, and the results are saved separately to ensure data integrity and prevent conflicts. Throughout the preprocessing process, logging is used to track the progress and any errors that occur. The log file provides a detailed record of the preprocessing steps, allowing for easy troubleshooting and analysis of the preprocessing pipeline.

**Face Detection**    The first step in the preprocessing pipeline is face detection. We employ the Dlib library, which provides an efficient face detection algorithm. The face detector scans each video frame and identifies the bounding boxes that enclose the faces.

**Face Alignment**    Once the faces are detected, the next step is face alignment. Face alignment refers to the process of transforming the faces in the images to a standardized pose. In our preprocessing script, we use facial landmarks to perform face alignment. We utilize the Dlib library, which provides a *pre-trained shape predictor model* that can effectively detect facial landmarks. Using the *shape predictor model*, we extract the facial landmarks for each detected face in the image. Specifically, we extract the landmarks for the eyes, nose, and mouth. These landmarks serve as reference points for aligning and cropping the face. To align the faces, we use an *affine transformation*, which is a linear mapping that preserves the shape of the face. The transformation is estimated based on the detected landmarks and a set of target landmarks, which define the desired position and size of the face. We apply the transformation to the original image to obtain the aligned face.

**Face Cropping**    After aligning the faces, the next step in the preprocessing pipeline is face cropping. To perform face cropping, we utilize the aligned faces obtained from the alignment step. To account for variations in face size and position, we introduce one parameter: **margin**. The margin parameter determines the amount of space around the aligned face that is included in the cropped image. Too large of a margin in the face cropping process can lead to the overfitting of the detection models to the contextual information surrounding the face, rather than focusing on the facial features themselves. This may result in the model relying more on irrelevant background details and thus reducing its generalization performance on unseen data. On the other hand, using too small of a margin in the face cropping process can lead to incomplete facial information being captured in the cropped face images. This occurs because a small margin restricts the region of interest to only the immediate vicinity of the aligned face. As a result, important facial features or parts of the face that extend beyond this limited region may be excluded from the cropped images. Therefore, there may exist a trade-off when choosing the margin parameter in the face-cropping process. In this paper, **we fix the margin to be 1.3 for all datasets** following the previous work [3]. **For the overall face cropping process**, we first calculate the bounding box of the aligned face region. The bounding box is then expanded by applying the margin parameter, which increases the size of the region of interest. Finally, we resize the expanded bounding box to the desired scale, resulting in a cropped face image with consistent dimensions (fixed with 256×256 in this paper).

**Landmark Extraction**    Extracting landmarks is an essential step in the preprocessing pipeline as it provides valuable information about facial structure and geometry. Landmarks are specific points on the face, such as the corners of the eyes, nose, and mouth, that serve as reference points for various

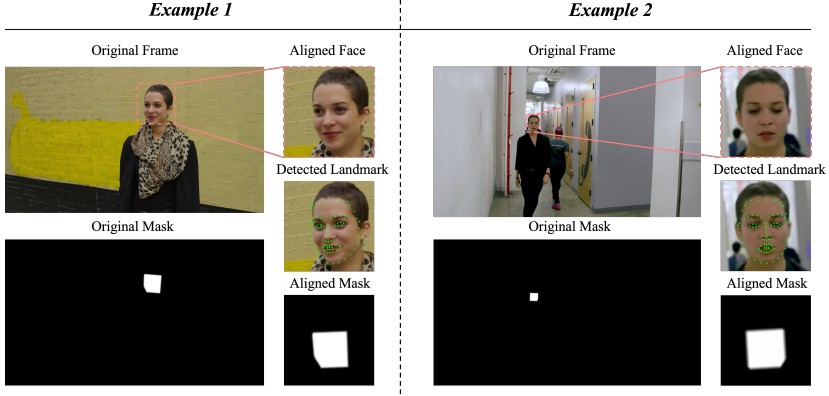

Figure 7: Illustration and visualization of the preprocessing procedure. We perform face detection and aligned cropping to the frame and its corresponding mask.

facial analysis algorithms. Several algorithms, such as Face X-ray [20], FWA [22], SLADD [3], *etc*, rely on landmarks to perform operations and analysis on facial images. By extracting landmarks during the preprocessing step, we aim to provide a comprehensive dataset that includes both the aligned face images and the corresponding landmark coordinates. Users can leverage landmarks to develop and train models without the need for additional face-detection steps during training, thereby reducing computational overhead and improving training speed.

**Mask Extraction (Optional)** In some deepfake datasets (*i.e.*, FaceForensics++ [33] and DFD [9]), an additional mask is provided, indicating the regions of the face that are manipulated or modified. Also, we see that there are several works that rely on mask data for the detection, *e.g.*, Multi-task [28], Face X-ray [20], M2TR [47], *etc*. Thus, if the dataset includes masks, our script also extracts and saves these masks. Since we have performed the face alignment and cropping operations in the previous steps, we need to do the same operations for the mask data. To extract masks, we utilize the additional video files provided by the authors that contain the mask information for each frame. Note these mask video files have the same frame count and frame rate as the original video. During the face cropping step, if a mask video file is provided, we extract the corresponding frames and masks for each video. The mask data is saved as a separate folder but with the same dictionary as the videos and frames. The masks can be used to identify specific areas of interest for further analysis or to train models that specifically focus on detecting manipulated regions.

**Frame Sampling** In the preprocessing pipeline, we incorporate frame sampling techniques to strike a balance between computational requirements and maintaining a diverse set of examples. This step aims to extract a subset of frames from each video in the dataset. The frame sampling process depends on the specified mode, which can be either **"fixed_num_frames"** or **"fixed_stride"**. **In the "fixed_num_frames" mode**, we extract a fixed number of frames from each video. This approach ensures that the resulting dataset contains a consistent number of frames for each video, regardless of the video's duration. By selecting a predetermined number of frames, we obtain a manageable dataset size that is suitable for subsequent analysis or model training. **In the "fixed_stride" mode**, we sample frames with a fixed stride. This means that we skip a certain number of frames between each frame that is selected. This approach allows us to capture frames at regular intervals throughout the video, providing a representative sampling of the temporal dynamics. By choosing an appropriate stride, we can control the density of the selected frames and adjust the amount of temporal information included in the dataset. **Frame sampling serves two primary purposes**. **Firstly**, it reduces the computational requirements for subsequent steps in the pipeline, such as face detection and alignment, by operating on a subset of frames rather than the entire video. This improves the overall efficiency of the preprocessing process, particularly when dealing with large-scale datasets. **Secondly**, frame sampling ensures that the resulting dataset maintains a diverse set of examples. By selecting frames at regular intervals or a fixed number of frames per video, we capture different facial expressions, poses, and actions exhibited by individuals. This diversity enhances the generalizability of models trained on the dataset, enabling them to handle a wide range of scenarios and variations encountered in real-world applications. **Note in this paper we only choose the "fixed_num_frames" mode**.

**Parallel Processing**   To improve the processing speed, we use parallel processing techniques. We leverage the *concurrent.futures library*, which provides a high-level interface for asynchronously executing callables. By using multiple processes, we can process multiple videos simultaneously, significantly reducing the overall processing time. The number of processes used is determined based on the CPU capabilities of the system. We assign one process per CPU core to maximize the utilization of available resources.

**Saving Processed Data**   After completing the preprocessing steps, we save the processed data for future use. The cropped face images, extracted landmarks, and masks (if available) are saved in a structured directory format. Each video is associated with a separate directory, containing the processed frames, landmarks, and masks (if applicable). This organization allows for efficient data retrieval and analysis during subsequent stages.

**Arrangement**   The process of rearranging the dataset structure is motivated by the need for a unified and convenient way to load different datasets. Each dataset typically has its own distinct structure and organization, making it hard and troublesome to handle them uniformly. This could involve writing separate input/output (I/O) code for each dataset, leading to duplication of effort and potential difficulties in managing the data.

To this end, we adopt a unified approach by organizing and managing the dataset information using a **JSON file**. This enables a standardized structure that subsequent algorithms and models can easily process. By leveraging the **JSON file** format, we provide a comprehensive and adaptable representation of the dataset, accommodating the specific requirements and characteristics of each dataset. The rearranged structure organizes the data in a hierarchical manner, grouping videos based on their labels and data splits (*i.e.*, train, test, validation). Each video is represented as a dictionary entry containing relevant metadata, including file paths, labels, compression levels (if applicable), *etc*. This unified representation facilitates streamlined dataset loading and handling, eliminating the need for dataset-specific I/O code.

The JSON file serves as a centralized repository of dataset information, providing a consistent and easily accessible format. Users can leverage existing code and tools to parse and analyze the JSON file, promoting reproducibility and facilitating collaborations across different datasets. Additionally, the JSON file simplifies the data preprocessing pipeline, reducing duplication of effort and enhancing the efficiency of subsequent data analysis and model training processes.

The whole process of data preprocessing and arrangement can be summarized in the following Algorithm. 1.

---

**Algorithm 1** Data Preprocessing and Arrangement

---

 1: **Input:** Video dataset
 2: **Output:** Preprocessed dataset with rearranged structure
 3: **Procedure:**
 4: Perform the following preprocessing steps for each video in the dataset:
 5:     Extract a subset of frames from each video using frame sampling techniques.
 6:     Detect faces in each video frame using the Dlib face detection algorithm.
 7:     Align and crop the faces based on the detected facial landmarks using *Dlib shape predictor model*.
 8:     (Optional) Extract and save masks for each aligned face if provided.
 9:     Extract landmarks for each detected face using *Dlib shape predictor model*.
10:     Save the processed face images, landmarks, and masks (if applicable) in separate folders.
11: Use parallel processing to speed up the overall processing time by processing multiple videos simultaneously.

12: Save the processed data in a structured directory format with a *JSON* file containing metadata.
13: **Return** the rearranged dataset structure with metadata stored in the *JSON* file.

---

**Configuration**   The provided config file contains settings for two different preprocessing tasks: "preprocess" and "rearrange". We will go through each section and explain the available settings and their advantages in this section.

For the **Preprocess**:

- dataset_name: This setting allows the user to specify the name of the dataset. Users can choose from a list of supported dataset names such as FaceForensics++ [33], Celeb-DF-v1 [23], Celeb-DF-v2 [23], DFDCP [11], DFDC [10], DeeperForensics-1.0 [17], and UADFV [21]. Each dataset has its own characteristics and purpose.

- dataset_root_path: This setting defines the root path where the dataset is located. Users need to provide the path to the dataset directory.

- comp: This setting is specific to the FaceForensics++ dataset and determines the compression level of the videos. Users can choose from "raw", "c23", or "c40". Different compression levels have different trade-offs between video quality and file size.

- mode: This setting determines the mode of preprocessing, either "fixed_num_frames" or "fixed_stride". In "fixed_num_frames" mode, users can specify the number of frames to extract from each video using the "num_frames" setting. In "fixed_stride" mode, users can specify the number of frames to skip between each frame extracted using the "stride" setting.

- stride: This setting is used when the mode is set to "fixed_stride". It determines the number of frames to skip between each frame extracted. A higher stride value will result in fewer extracted frames.

- num_frames: This setting is used when the mode is set to "fixed_num_frames". It specifies the number of frames to extract from each video. Extracting a fixed number of frames allows for consistent and manageable data sizes.

For the **Arrangement**:

- dataset_name: This setting allows users to specify the name of the dataset users want to rearrange.

- dataset_root_path: This setting defines the root path where the dataset is located.

- output_file_path: This setting specifies the path where the output JSON file will be saved. The JSON file contains information about the rearranged dataset.

- comp: This setting is specific to the FaceForensics++ dataset and determines the compression level of the videos. Users can choose from "raw", "c23", or "c40".

- perturbation: This setting is specific to the DeeperForensics-1.0 dataset and allows users to select different levels of perturbations to apply to the dataset. There are options such as "end_to_end", "end_to_end_level_1", "end_to_end_mix_2_distortions", *etc*.

Dataset rearrangement is specifically designed for rearranging datasets. It provides the flexibility to modify and rearrange the dataset according to specific needs. The script generates a JSON file that contains information about the rearranged dataset. This file can be used for further analysis or as input to other scripts or models. By using this config file, users can easily customize the preprocessing and rearrangement tasks to suit their specific dataset and requirements. The flexibility offered by this file enables efficient and consistent preprocessing of various deepfake datasets.

### A.2 Details of Algorithms Implementation and Visualizations

**Algorithms Implementation**    In addition to the basic information in Tab. 2 of the main manuscript, here we describe the general idea of the 15 implemented detection algorithms in the *DeepfakeBench*, as follows.

1) **Meso4** [1]: is a CNN-based deepfake detection method targeting the mesoscopic properties of images. The model is trained on unpublished deepfake datasets collected by the authors. We evaluate two variants of MesoNet, namely, Meso4 and MesoIncep. Meso4 uses conventional convolutional layers.

2) **MesoIncep** [1]: this detector, similar to Meso4, utilizes a designed CNN architecture and is also implemented in the MesoNet repository. Note that MesoIncep is based on the more sophisticated Inception modules [39].

3) **CNN-Aug** [48]: detects GAN-generated images using a ResNet [16] with widely-used augmentations. In the *DeepfakeBench*, we employ a ResNet-34 [16] with JPEG compression and Gaussian

blurring augmentations, *etc.* The effect of augmentations we used in this work has been explored in Sec. 4 in the main paper. The specific settings of the augmentations can be found in the following section in Sec. A.3.

4) **EfficientNet-B4** [40]: is Based on the EfficientNet architecture [40]. We find that many detectors utilize this architecture as their basic backbone for feature extraction (*e.g.*, SBIs [37], multi-attention [52], *etc*). Also, as we implement this framework in our benchmark, we can compare the performance of different basic architectures and find the improvement bring by only the architecture.

5) **Xception** [33]: corresponds to a deepfake detection method based on the XceptionNet model [5] trained on the FaceForensics++ dataset [33]. There are three variants of Xception, namely, Xception-raw, Xceptionc23, and Xception-c40: Xception-raw is trained on raw videos, while Xception-c23 and Xception-c40 are trained on H.264 videos with medium (23) and high degrees (40) of compression, respectively.

6) **Capsule** [29]: uses capsule structures [34] based on a VGG19 [38] network as the backbone architecture for deepfake classification. This model is originally trained on the FaceForensics++ dataset [33].

7) **DSP-FWA** [22]: detects deepfake videos using a ResNet-50 [16] to expose the face-warping artifacts introduced by the resizing and interpolation operations in the basic deepfake maker algorithm. This model is trained on self-collected face images. In the original paper, DSP-FWA further improves the FWA algorithm by including a spatial pyramid pooling (SPP) module [15] to better handle the variations in the resolutions of the original target faces. Note that in the *DeepfakeBench*, we do not adopt the SPP module since we try to use the same architecture (backbone) for each detector so that we can find the actually effective technologies toward deepfake detection. Instead, we use the standard Xception for this detection as other detectors. However, we utilize the multi-scale strategy in the dynamic forgery data generation process to obtain different scale faces blending (the scale parameters we set are $[0.2, 0.3, 0.4, 0.5, 0.6, 0.7, 0.8]$). Following its paper, we first align the face image to multiple scales and randomly select an aligned image. We visualize blending examples to show that our implementation can achieve similar forgery samples as the original paper (see Sec. A.2).

8) **Face X-ray** [20]: uses blended artifacts in forgeries to improve generalization ability to detect unseen forgeries. In this work, following the original paper, we train an HRNet [46] both with constructed blended images and fake samples from the considered datasets (FaceForensics++ [33] in our main experiments). Note that the code for this detector is not publicly available, we re-implement it carefully following the instructions and settings in the original paper. We visualize blending examples to show that our implementation can achieve similar forgery samples as the original paper (see Sec. A.2).

9) **FFD** [6]: applies an attention mechanism to detect and localize manipulation regions. The author proposes two types of attention-based layers, named manipulation appearance model and direct regression, to guide the network to focus on discriminative regions. Meanwhile, three types of loss functions are proposed to supervise the learning progress. In our implementation, we adopt the Xception [5] as the backbone and direct regression as the attention-based layer to train the model.

10) **CORE** [30]: explicitly constrains the consistency of different representations. Different representations are first captured with different augmentations, and then the cosine distance of the representations is regularized to enhance consistency. This detector utilizes the Xception backbone [5].

11) **RECCE** [2]: constructs a graph over encoder and decoder features in a multi-scale manner. It further utilizes the reconstruction differences as the forgery traces on the graph output as a guide to the final representation, which is fed into a classifier for forgery detection. End-to-end optimization for reconstruction and classification learning.

12) **UCF** [50]: introduces a multi-task disentanglement framework to address two main challenges that contribute to the generalization problem in deepfake detection: overfitting to irrelevant features and overfitting to method-specific textures. By uncovering common features, the framework aims to enhance the generalization ability of the model. This detector utilizes the Xception backbone [5]. The code for this detector is not publicly available, we re-implement it carefully following the instructions and settings in the original paper.

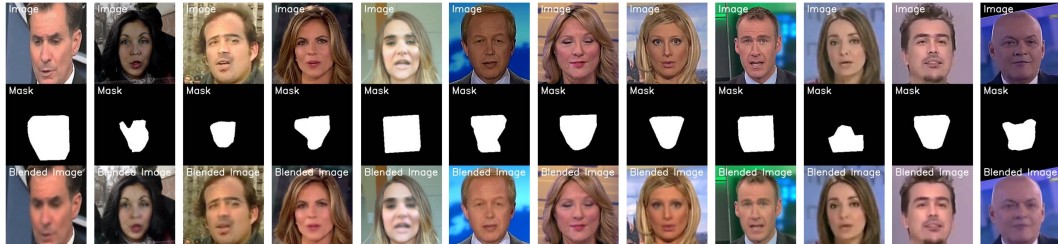

Figure 8: Illustration and visualization of the DSP-FWA algorithm. We use the data from FaceForensics++ [33] and apply some augmentations to the source image, as well as the blending image.

13) **F3Net** [32]: uses cross-attention two-stream networks to collaboratively learn frequency-aware clues from two branches: FAD and LFS, where the FAD module partitions the input image in the frequency domain based on learnable frequency bands and represents the image with frequency-aware components to learn forgery patterns through frequency-aware image decomposition, and the LFS module extracts localized frequency statistics to describe statistical discrepancies between real and fake faces, allowing for effective mining through CNNs and revealing unusual statistics of forgery images at each frequency band while sharing the structure of natural images. This detector utilizes the Xception backbone [5]. The code for this detector is not publicly available, we re-implement it carefully following the instructions and settings in the original paper.

14) **SPSL** [26]: combines spatial image and phase spectrum to capture the up-sampling artifacts of face forgery to improve the transferability (generalization ability), for face forgery detection. This paper theoretically analyzes the validity of utilizing the phase spectrum. Moreover, this paper notices that local texture information is more crucial than high-level semantic information for face forgery detection. This detector utilizes the Xception backbone [5]. The code for this detector is not publicly available, we re-implement it carefully following the instructions and settings in the original paper.

15) **SRM** [27]: extracts high-frequency noise features and fuses two different representations from RGB and frequency domains to improve the generalization ability. This detector utilizes the Xception architecture [6]. This detector utilizes the Xception backbone [5]. The code for this detector is not publicly available, we re-implement it carefully following the instructions and settings in the original paper.

**Visualizations**  We implement all 15 detectors mentioned above. However, not all of them have publicly available code, so we implement some of them ourselves following the settings and instructions provided in the original papers. This allowed us to verify the correctness of our implementation and gain a better understanding of these detectors. To further assess the performance and behavior of the detectors, we conduct visualizations of the results for 2 specific detectors: DSP-FWA [22] and Face X-ray [20].

1) **DSP-FWA**: Note that the official code for DSP-FWA does not include the training code or the code for dynamically generating forgery data using self-blending in each iteration during training. To this end, we make use of certain parts of the code provided in the official repository and implement the training process and forgery data generation ourselves. In our implementation of DSP-FWA, we use the Xception network [5] as the backbone. This choice is to ensure consistency in the benchmark by using the same backbone network across different detectors. By doing so, we could focus solely on evaluating the algorithmic performance of DSP-FWA itself. By incorporating our own implementation of the training process and forgery data generation, we are able to overcome the absence of these components in the official code. This allows us to thoroughly evaluate DSP-FWA and ensure a fair comparison with other detectors in our benchmark. We visualize the original images, blending masks, and blending images in Fig. 8.

2) **Face X-ray**: Note that the official code for Face X-ray is not available. So we re-implement the data manipulation and training process carefully following the instructions of the original paper. The visualizations can be seen in Fig. 10.

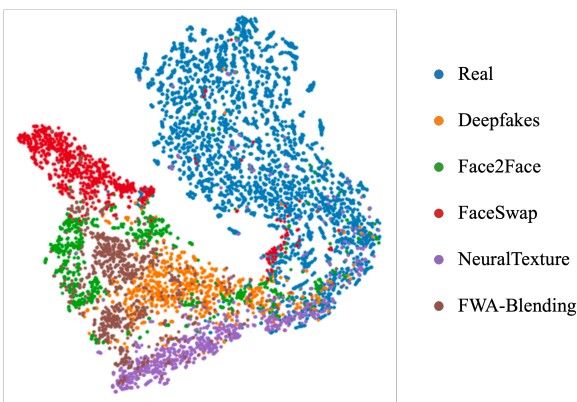

Figure 9: t-SNE visualization of FWA-generated data. By assigning distinct labels to various forgeries, we enhance the clarity of their representation within the feature space.

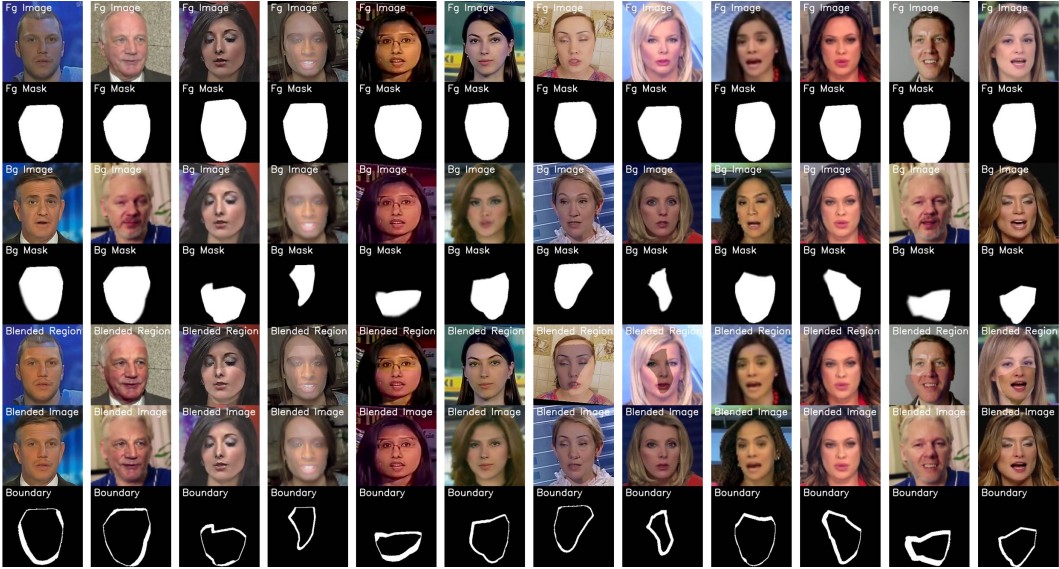

Figure 10: Illustration and visualization of the Face X-ray algorithm. We use the data from Face-Forensics++ [33] and apply some augmentations to the source image, as well as the blending image.

Furthermore, we conduct a t-SNE analysis for FWA, visualizing labels in the feature space. Our findings suggest that images generated through blending technology (new data generated by FWA) exhibit distinctiveness, distancing them from images generated by alternative manipulation methodologies. This characteristic enlarges the forgery space, culminating in enhanced generalization capabilities.

### A.3 Training Details and Full Experimental Results

***Datasets*** Our benchmark currently incorporates a collection of 9 widely recognized and extensively used datasets in the realm of deepfake forensics: FaceForensics++ (FF++) [33], CelebDF-v1 [23], CelebDF-v2 [23], DeepFakeDetection (DFD) [9], DeepFake Detection Challenge Preview (DFDC-P) [11], DeepFake Detection Challenge (DFDC) [10], UADFV [21], FaceShifter [19], and DeeperForensics-1.0 (DF-1.0) [17]. The detailed descriptions of each dataset are presented in Tab. 5.

The dataset splitting for different datasets used in deepfake detection is described as follows:

1) **FaceForensics++ (FF++):** The FF++ dataset is divided into several subsets, including FF-DF, FF-F2F, FF-FS, FF-NT, and FF-all. Each subset corresponds to a combination of deepfake and real videos from YouTube. In the real dataset, the data is duplicated and split into three sets: train, test,

| Dataset | Real Videos | Fake Videos | Total Videos | Rights Cleared | Total Subjects | Synthesis Methods | Perturbations | Download Link |
|---|---|---|---|---|---|---|---|---|
| FF++ [33] | 1000 | 4000 | 5000 | NO | N/A | 4 | 2 | Hyper-link |
| FaceShifter [19] | 1000 | 1000 | 2000 | NO | N/A | 1 | - | Hyper-link |
| DFD [9] | 363 | 3000 | 3363 | YES | 28 | 5 | - | Hyper-link |
| DFDC-P [11] | 1131 | 4119 | 5250 | YES | 66 | 2 | 3 | Hyper-link |
| DFDC [10] | 23,654 | 104,500 | 128,154 | YES | 960 | 8 | 19 | Hyper-link |
| CelebDF-v1 [23] | 408 | 795 | 1203 | NO | N/A | 1 | - | Hyper-link |
| CelebDF-v2 [23] | 590 | 5639 | 6229 | NO | 59 | 1 | - | Hyper-link |
| DF-1.0 [17] | 50,000 | 10,000 | 60,000 | YES | 100 | 1 | 7 | Hyper-link |
| UADFV [21] | 49 | 49 | 98 | NO | 49 | 1 | - | Hyper-link |

Table 5: *Summary of the datasets used for deepfake detection. The table provides information on the number of real and fake videos, the total number of videos, whether rights have been cleared, the number of agreeing subjects, the total number of subjects, the number of synthesis methods, and the number of perturbations.*

and validation. For the fake dataset, the train, test, and validation splits are determined based on the information provided in the corresponding JSON files used in the arrangement process (see Sec. A.1). Masks are also included in the dataset.

2) **DeepFakeDetection:** Since the dataset does not have the official splitting, the fake and real data are duplicated and split into the train, test, and validation sets in our benchmark. Masks are included in this dataset as well.

3) **FaceShifter:** The real data is duplicated and split into the train, test, and validation sets, similar to the FF++ dataset. The train, test, and validation splits for the fake dataset are determined using the FF++ JSON files used in the arrangement process.

4) **Celeb-DF-v1/v2:** All the real and fake videos are used as the training dataset, and a subset of real and fake videos is selected as the test dataset based on a text file provided by the author. The validation set is set to be the same as the test set.

5) **DFDCP:** The dataset contains real videos and fake videos generated by two different methods: method A and method B. The train and test splits are determined based on the given method. The validation set is set to be the same as the test set.

6) **DFDC:** The train and test splits are determined based on the given method, similar to DFDCP. The validation set is set to be the same as the test set.

7) **DeeperForensics-1.0:** The dataset includes various perturbation methods in the fake data subset. One perturbation method is considered a separate category of fake videos. In the fake dataset, the train, test, and validation splits are determined based on the provided text file. The real dataset is duplicated and split into train, test, and validation sets.

8) **UADFV:** The strategy used for the UADFV dataset involves duplicating the real and fake parts of the dataset three times to create the train, test, and validation sets.

**Experimental Setup**   In the training module, we utilize the Adam optimization algorithm with a learning rate of 0.0002. The batch size is set to 32 for most experiments. However, for the DSP-FWA [22] and Face X-ray [20] detectors, the batch size is adjusted to 16 due to the input data being pairs. Specifically, for DSP-FWA and Face X-ray, which generate forgery images dynamically during training, the input size is doubled.

For the naive detectors (*e.g.*, ResNet, Xception, and EfficientNet), we employ their official models, initializing the parameters through pre-training on the ImageNet. The pre-trained backbones from ImageNet are used to initialize the remaining weights. However, Meso4 [1] and MesoIncep [1] do not have pre-training weights in ImageNet, so pre-training is not utilized for them. The effect of pre-training is evaluated in Sec. 4.3 of the main paper.

Regarding evaluation, we compute the average value of the top-3 metrics, such as the average top-3 Area Under the Curve (AUC), as our primary evaluation metric. Additionally, we report the top-1 results. Other widely used metrics, including Average Precision (AP) and Equal Error Rate (EER), are also computed and presented in the following sections. Furthermore, it is important to note that the validation set is not utilized in our experiments. Following previous works [3, 4], we adopt the practice of selecting the model that achieves the highest performance on the test set rather than the validation set for final evaluation.

To ensure fair and consistent evaluation, all experiments are conducted in a standardized environment using the NVIDIA A100 GPU. More software library dependencies can be seen on our GitHub website (https://github.com/SCLBD/DeepfakeBench).

**Data Augmentation**   Our benchmark utilizes a series of widely used data augmentation methods for image processing. We describe each augmentation method as follows:

1) **Horizontal Flip:** This augmentation randomly flips the image horizontally with a probability of 0.5, simulating mirror images.

2) **Rotation:** This augmentation randomly rotates the image within a range of -10 to 10 degrees with a probability of 0.5. By applying random rotations, it introduces diversity in object orientations, making the model more robust to different angles and orientations.

3) **Isotropic Resize:** This augmentation resizes the image while maintaining isotropy, ensuring that the aspect ratio of the image is preserved. It randomly selects one interpolation method (INTER_AREA, INTER_CUBIC, or INTER_LINEAR) for resizing. The maximum side length is determined by the configured value. Isotropic resizing is particularly useful when dealing with objects that have varying scales and proportions, allowing the model to learn from different object sizes and maintain the aspect ratio of the objects.

4) **Random Brightness and Contrast:** This augmentation randomly adjusts the brightness and contrast of the image with a probability of 0.5. By applying random brightness and contrast variations, it introduces changes in the illumination and contrast levels of the images. This helps the model generalize better to different lighting conditions and improves its robustness to variations in brightness and contrast.

5) **FancyPCA:** This augmentation applies the FancyPCA algorithm with a probability of 0.5. FancyPCA performs Principal Component Analysis (PCA) on the pixel values of the image and perturbs the components to introduce color variations. By altering the principal components of the image, it can change the color distribution, leading to more diverse training samples.

6) **Hue Saturation Value (HSV) Adjustment:** This augmentation randomly adjusts the hue, saturation, and value of the image. While the probability is not specified in the code snippet, it allows for variations in the color representation of the images. Adjusting the hue changes the overall color tone, saturation controls the intensity of colors, and value adjusts the brightness.

7) **Image Compression:** This augmentation applies image compression with a probability of 0.5. It reduces the quality of the image by compressing it. The lower and upper limits, set to 40 and 100 respectively, control the compression quality. Image compression introduces artifacts and reduces the image quality, simulating real-world scenarios where images may be of lower quality or have compression artifacts. This augmentation helps the model learn to handle such variations and improves its robustness in practical applications.

**Full Experimental Results**

**Overview**   In the main paper, our focus is on presenting the experimental results obtained from selecting the models that achieve the highest performance on each individual testing dataset. The primary metric utilized for evaluation in the main paper is the Area Under the Curve (AUC). In order to provide a more comprehensive view of our experimental results, we present the complete set of results here. We have incorporated three different widely utilized metrics for assessment: AUC, Average Precision (AP), and Equal Error Rate (EER). These metrics are dynamically recorded throughout the training process as part of our benchmark. Additionally, we have stored the prediction results along with their corresponding labels, which facilitates the computation of additional metrics. In this paper, we compare the 3 aforementioned metrics as a means to compare the performance of the 15 detectors across the 14 testing datasets.

**Comprehensive Metrics**   In addition to saving the best-performing model throughout the training process, we also save the last model to evaluate its performance at the completion of all training epochs. This allows us to assess the models' effectiveness after undergoing the entire training duration. Furthermore, by recording the predictions and corresponding labels, we are able to calculate additional metrics such as Precision and Recall, in addition to the previously mentioned metrics.

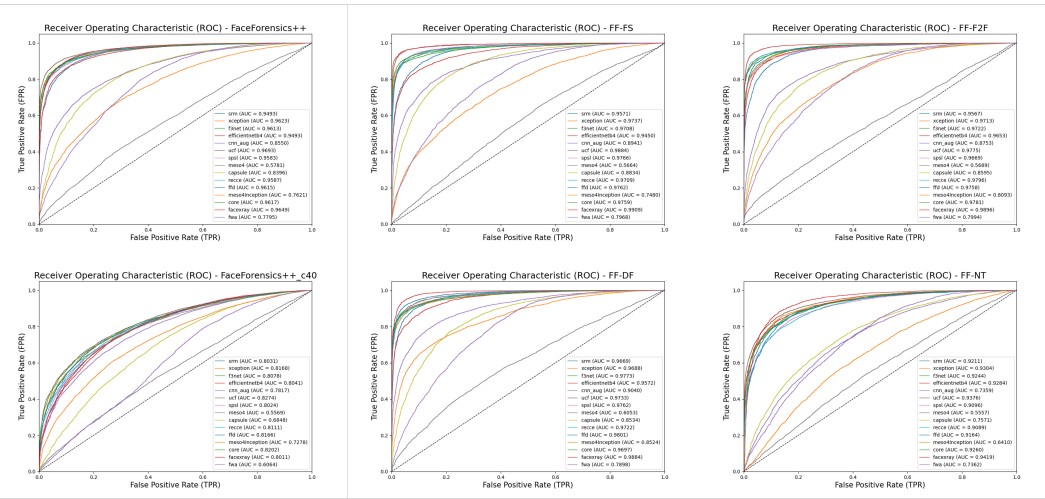

Figure 11: Illustration and visualization of within-dataset evaluation. We draw the ROC-AUC curve using the models at the last trained epoch.

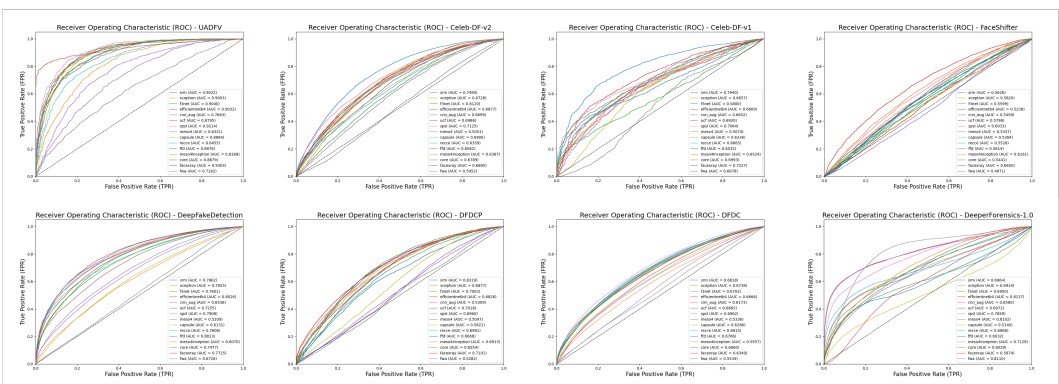

Figure 12: Illustration and visualization of cross-dataset evaluation. We draw the ROC-AUC curve using the models at the last trained epoch.

Here, we present the ROC-AUC curve and Precision-Recall curve for all detectors. These detectors are trained on the FF++ (c23) dataset and evaluated on a total of 14 testing datasets, encompassing both within-dataset and cross-dataset evaluations (see Fig. 11, Fig. 13, Fig. 12, Fig. 14). These visualizations provide a more comprehensive understanding of the experimental outcomes, allowing for a more detailed analysis of the detectors' performance. Moreover, as a benchmark, our proposed approach facilitates the computation of additional evaluation metrics based on user requirements, thereby demonstrating the convenience and versatility of our benchmarking framework.

**Full Testing Results During the Training Process**   To facilitate the monitoring of model performance during the training process, we utilize TensorBoard to record various metrics. These metrics include training loss, training accuracy, AUC, AP, and EER, as well as testing loss and testing metrics (AUC, AP, EER). By visualizing these metrics, users gain insight into the performance trends during training, enabling them to debug issues and optimize parameters as needed.

In this section, we present visualizations of testing metrics plotted against the training steps. The metrics of interest include AUC, AP, and EER, which provide a comprehensive assessment of the detectors' performance across different datasets (see Fig. 16, Fig. 19, Fig. 15, Fig. 18, Fig. 17, Fig. 20). By comparing the curves, we can analyze the relative performance of the detectors using different evaluation metrics.

Furthermore, we can observe the stability of the testing results. Some detectors may exhibit volatility and lack stability in their metrics, which introduces uncertainty. In such cases, while the overall

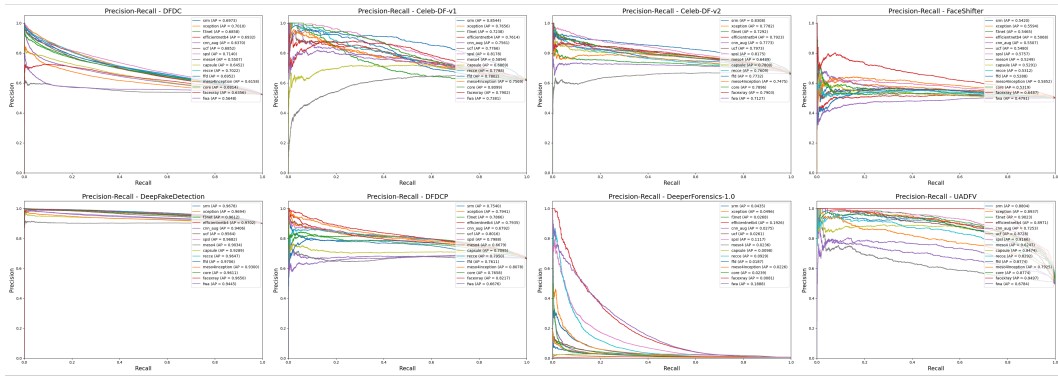

Figure 13: Illustration and visualization of within-dataset evaluation. We draw the Precision-Recall curve using the models at the last trained epoch.

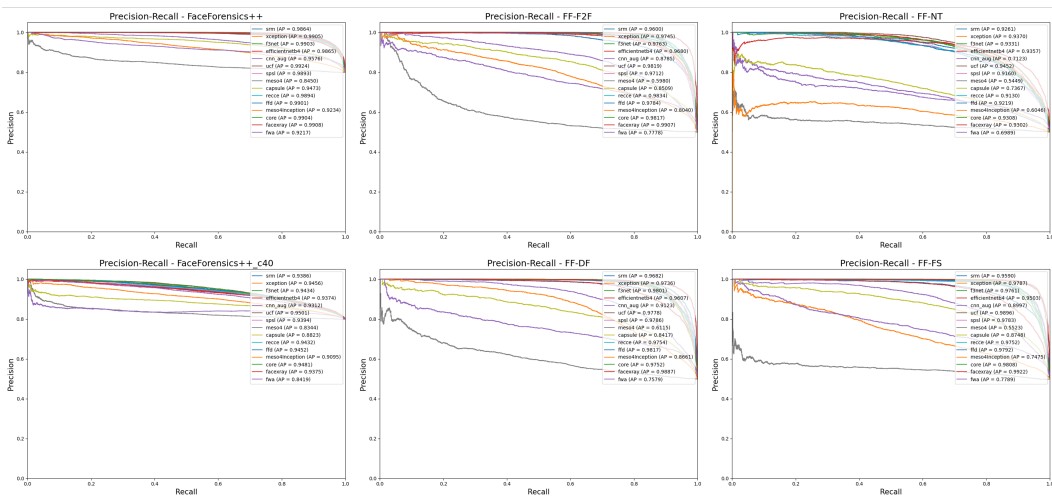

Figure 14: Illustration and visualization of cross-dataset evaluation. We draw the Precision-Recall curve using the models at the last trained epoch.

results may not be consistently good, there may be instances where individual metrics perform well. To address this, we adopt an average-based approach, computing the average values for each testing metric to determine the final results (Top-3). By examining the provided figures, we can also discern the stability of each detector's performance.

Note that due to the differing training batch sizes of DSP-FWA and Face X-ray detectors compared to the other 13 detectors, we visualize them separately. This distinction allows for a more clear comparison within their respective groups.

## A.4 Other Analysis Results

**Artifacts of deepfake forgeries in Frequency** Inspired by [48], we adopt a similar approach to visualize the average frequency spectra of each dataset. The purpose is to examine the artifacts generated by deepfake forgeries. Our methodology involves computing the average frequency spectrum of a selected set of images, specifically 2000 randomly sampled images. To mitigate computational complexity, a random subset of both real and fake images is chosen for analysis. The process begins by converting the images to grayscale and applying a high-pass filter. Subsequently, a Fourier transform is performed, with the zero frequency component shifted to the center of the spectrum. Finally, the spectra are summed and averaged to obtain the final result.

The resulting visualization comprises three subplots for each deepfake forgery. The first subplot illustrates the average spectrum of the real image, the second subplot represents the average spectrum

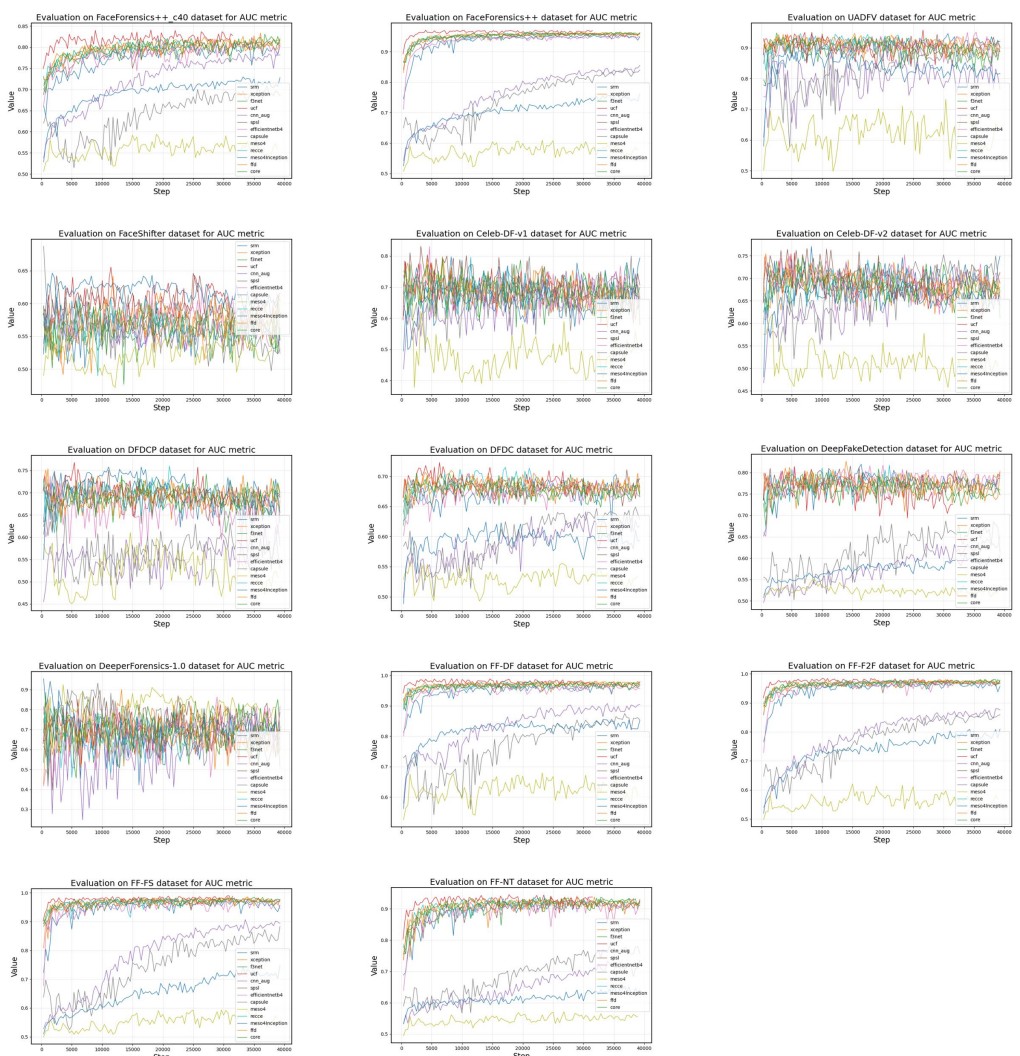

Figure 15: Illustration and visualization of all testing results during the training process. The metric is AUC. We compare 13 detectors (except for the Face X-ray and DSP-FWA) on different datasets using the AUC metric.

of the fake image, and the third subplot showcases the difference between the spectra of the real and fake images.

Our findings align with those reported in [48]. We observe that deepfake forgeries do not exhibit obvious artifacts, as observed in other images generated by GANs. This consistency with the findings in [48] can be attributed to the various pre-processing and post-processing steps involved in the creation of deepfake images. These steps, which include resizing, blending, and MPEG compression of the synthesized face region, introduce perturbations in the low-level image statistics. As a result, the frequency patterns may not emerge distinctly in our visualization method.

**Visualizations of GAN-generated and diffusion-generated artifacts in Frequency**  Following the similar process in Sec. A.4, we also visualize the artifacts generated by GANs and diffusion models. Specifically, we utilize the GenImage dataset [54] and apply the frequency analysis tool in our benchmark for analysis. The visualizations are shown in Fig. 22. This analysis has unearthed intriguing observations specific to diffusion-generated images when contrasted with GAN-generated images. Particularly, diffusion-generated images exhibit fewer artifacts, while GAN-generated images display a noticeable checkerboard pattern of artifacts.

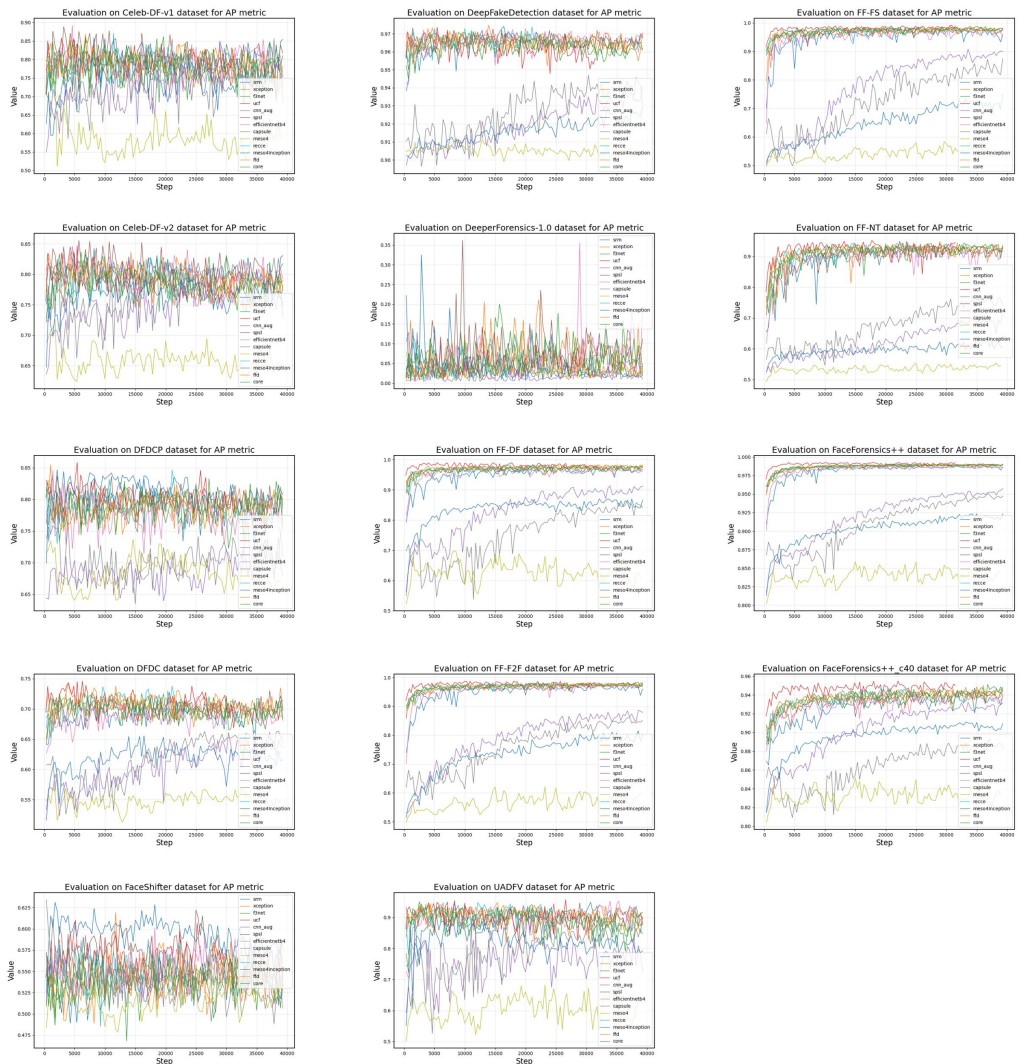

Figure 16: Illustration and visualization of all testing results during the training process. The metric is AP. We compare 13 detectors (except for the Face X-ray and DSP-FWA) on different datasets using the AP metric.

**Cross-data evaluation and the importance of phase spectrum**  In Tab. 3 of the manuscript, we highlight the SPSL detector, which achieves an impressive average score of 78.75% in cross-domain evaluation. A distinctive feature of SPSL compared to Xception is the incorporation of the phase spectrum, which is concatenated with the spatial image in the channel dimension. As mentioned in the original SPSL paper, the phase spectrum can capture up-sampling artifacts present in many forgery processes. Motivated by this finding, we explore the potential benefits of incorporating the phase spectrum feature into blending-based detectors. We hypothesize this would enhance performance in both cross-data and cross-manipulation evaluations.

- **Cross-data evaluation:** To validate our hypothesis, we first conduct an experiment in which we integrate the spectrum feature into the FWA detector, resulting in an improved FWA (iFWA). The experimental results, summarized in Tab 6, show a significant improvement achieved by iFWA (from 73.16% to 80.35% in average AUC).

- **Cross-manipulation evaluation:** Second, we conduct experiments and show the cross-manipulation outcomes achieved by iFWA in Tab. 7. These visualizations serve to strengthen our argument about the consistent performance of iFWA.

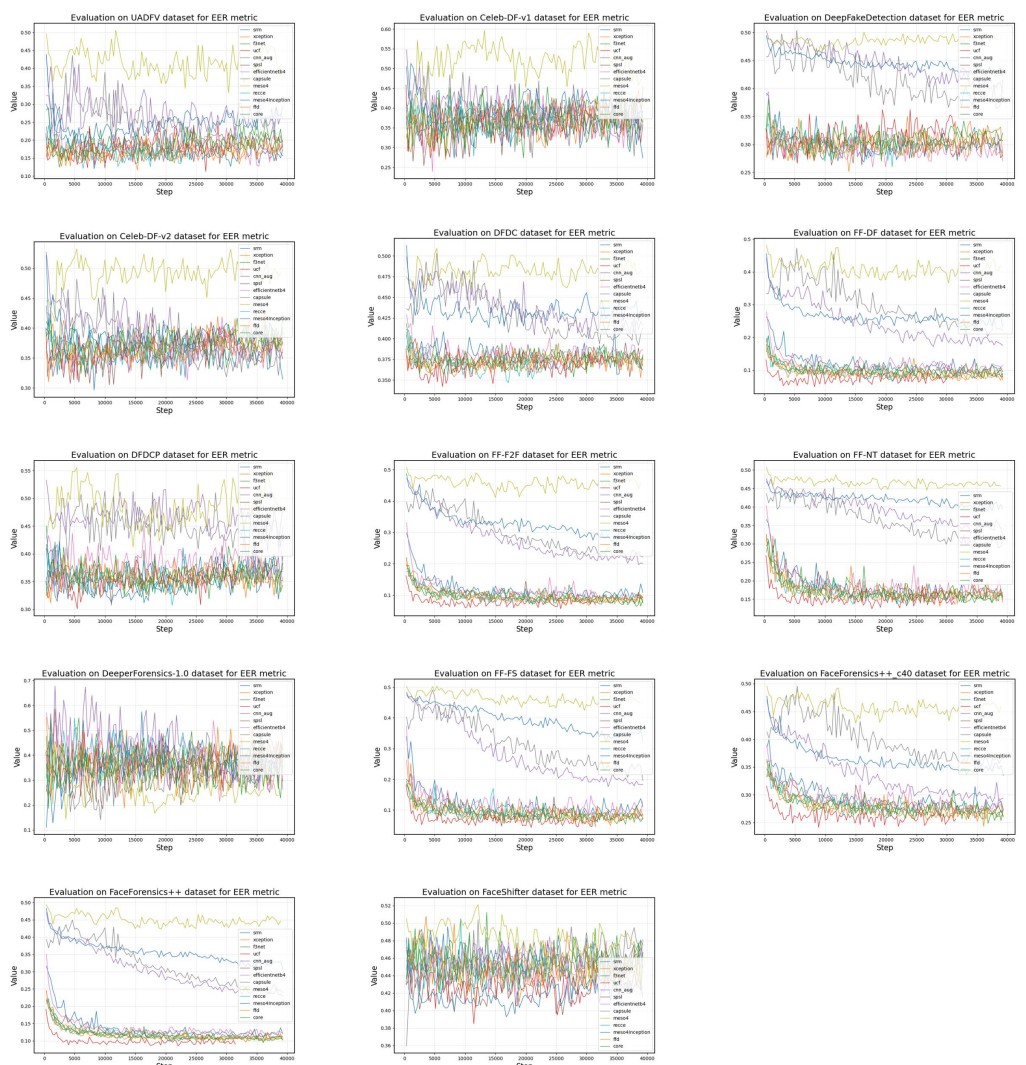

Figure 17: Illustration and visualization of all testing results during the training process. The metric is EER. We compare 13 detectors (except for the Face X-ray and DSP-FWA) on different datasets using the EER metric.

These two analyses and experimental validations aim to explain the phenomena observed in our evaluations, ensuring our experimental evaluations are not only fair and comprehensive, but also insightful.

| Model | FF++_c23 | FF++_c40 | CDF-v2 | DFDCP | DFD | Average |
|---|---|---|---|---|---|---|
| FWA | 0.8765 | 0.7357 | 0.6680 | 0.6375 | 0.7403 | 0.7316 |
| iFWA | 0.9557 | 0.7496 | 0.7612 | 0.7104 | 0.8408 | 0.8035 |
| Improvement | +7.92% | +1.39% | +9.32% | +7.29% | +10.05% | +7.19% |

Table 6: *Cross-data evaluation between iFWA (with the spectrum feature) and FWA (without the spectrum feature). The models are trained on FF++_c23 and tested on other datasets. The metric is the frame-level AUC.*

**Why do Naive detectors work can perform as well as more advanced in certain settings?** From results in Tab. 3, we have observed that some Naive detectors (*i.e.*, Xception and EfficientNetB4) can exhibit competitive performance compared to more complex methods, which might be surprising given the advancements in the field. We then explain this phenomenon from the following aspects.

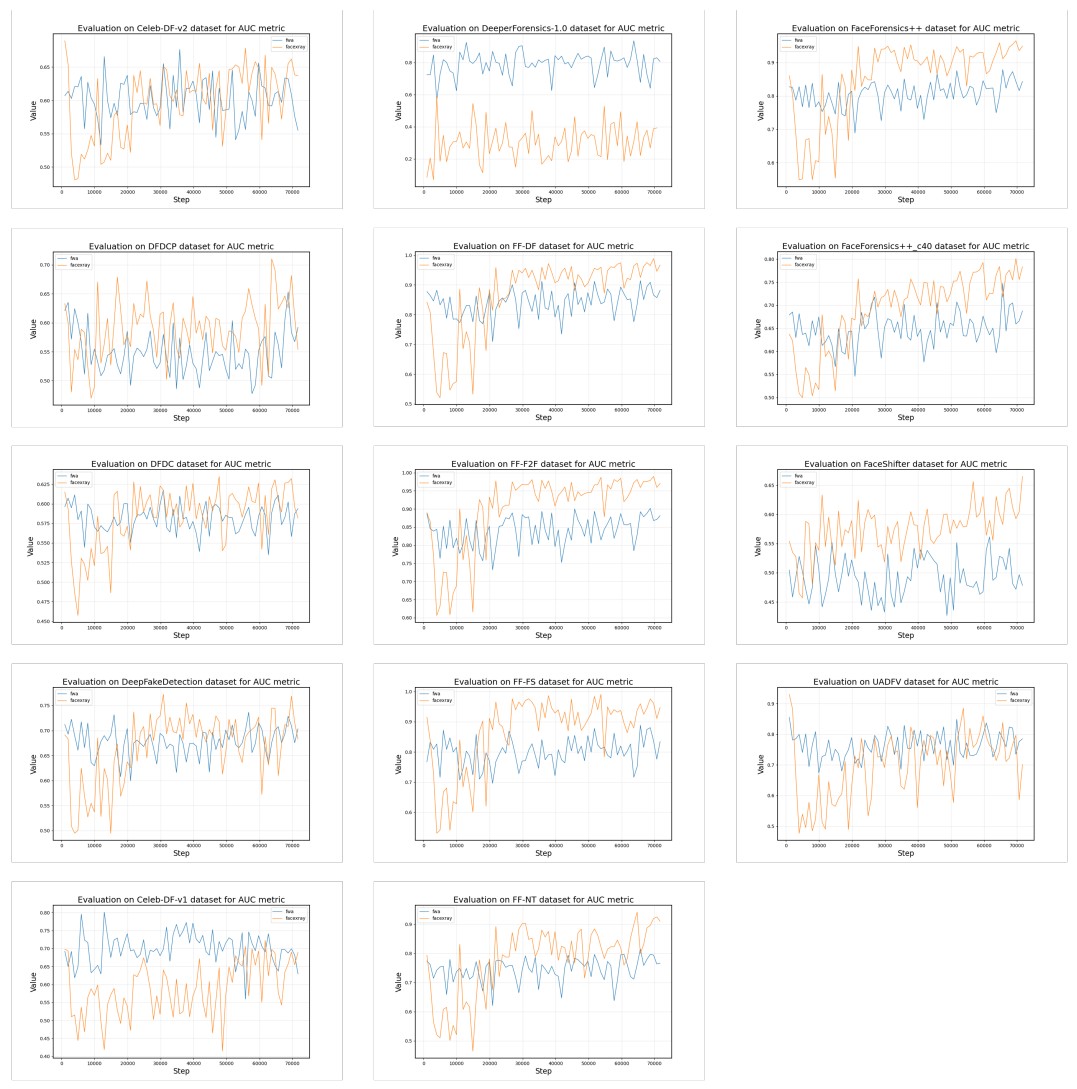

Figure 18: Illustration and visualization of all testing results during the training process. The metric is AUC. We compare Face X-ray and DSP-FWA on different datasets using the AUC metric.

| Model | Training | FF-DF | FF-F2F | FF-FS | FF-NT | Average |
|---|---|---|---|---|---|---|
| FWA | FF-DF | 0.90 | 0.91 | 0.92 | 0.90 | 0.91 |
| iFWA | FF-DF | 0.97 | 0.97 | 0.98 | 0.90 | 0.96 |
| Improvement | - | +7% | +6% | +6% | +0% | +5% |

Table 7: *Cross-manipulation evaluation between iFWA (with the spectrum feature) and FWA (without the spectrum feature). The models are trained on FF-DF and tested on other forgeries in FF++_c23. The metric is the frame-level AUC.*

**First**, Naive detectors, despite their simplicity, may have inherent strengths that are yet to be fully understood and harnessed. However, few previous studies have deeply explored the capabilities of these baseline methods or identified the conditions under which they can be particularly effective. **Second**, previous works have shown that some strategies or tricks could bolster the performance of Naive detectors, *e.g.*, pre-training or data augmentation. To this end, we conduct an experiment to compare the performance of the Naive detector and the complex one under the conditions with or without tricks. By adding the tricks, we find the gap between the Naive detector and complex detector

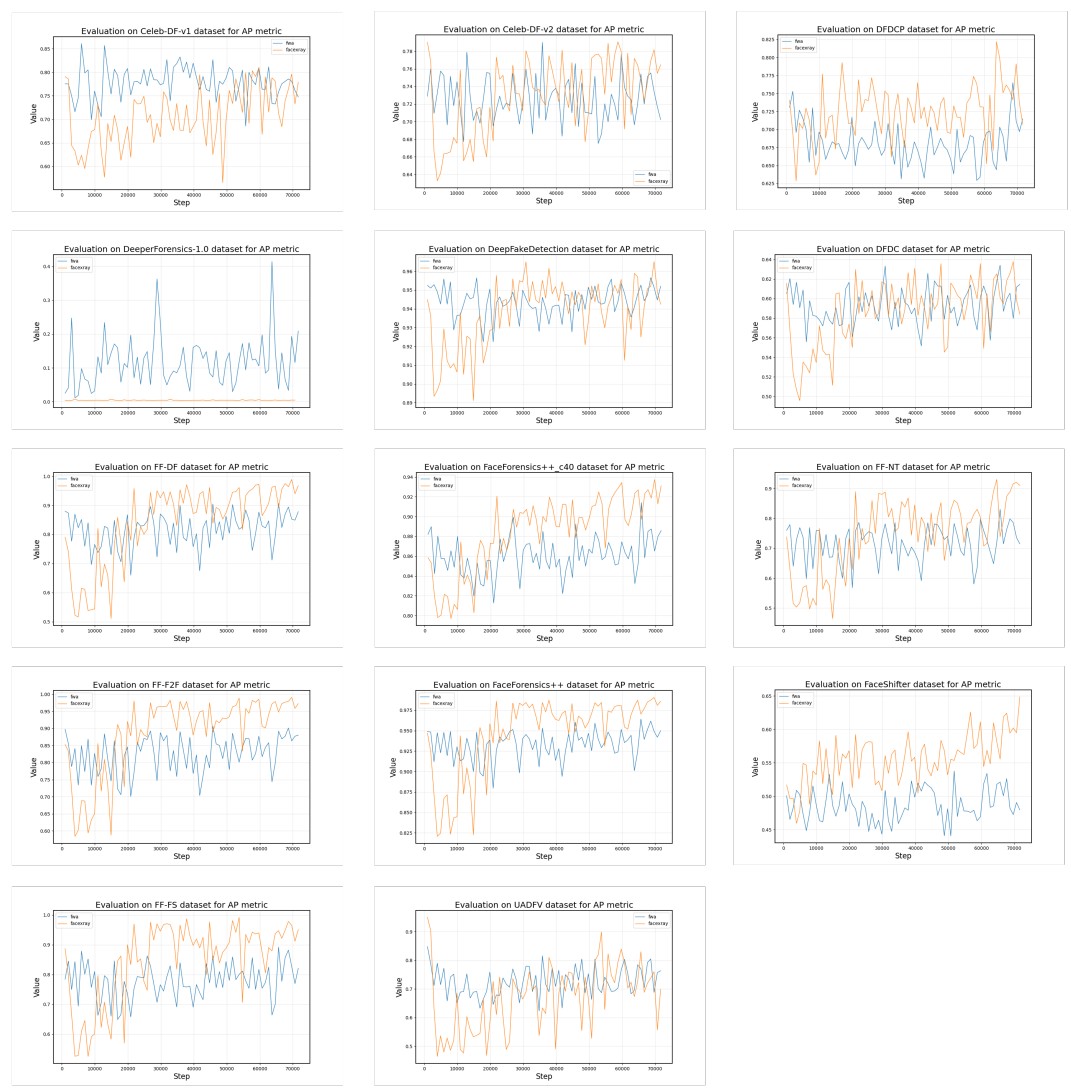

Figure 19: Illustration and visualization of all testing results during the training process. The metric is AP. We compare Face X-ray and DSP-FWA on different datasets using the AP metric.

| Model | Number of Layers | Number of Parameters |
|---|---|---|
| Xception | 71 | 22.9M |
| ResNet 34 | 34 | 21.8M |
| EfficientNet-B4 | $\sim$75 | 19M |

Table 8: *Summary of the statistics for Xception, ResNet 34, and EfficientNet-B4.*

is reduced (see Tab. 10). Showing that the performance of the Naive detector is effectively mined by using these tricks.

Apart from these two aspects, we also find that the problem of consistency in experimental procedures and evaluation metrics is also notable. It is worth noting that comparison methodologies can vary across studies, and directly adopting results from prior papers could sometimes lead to discrepancies due to differences in experimental conditions and evaluation metrics. For instance, current studies often directly cite the results of Xception from the original paper [33], but different training settings used in different works can inevitably result in disparities.

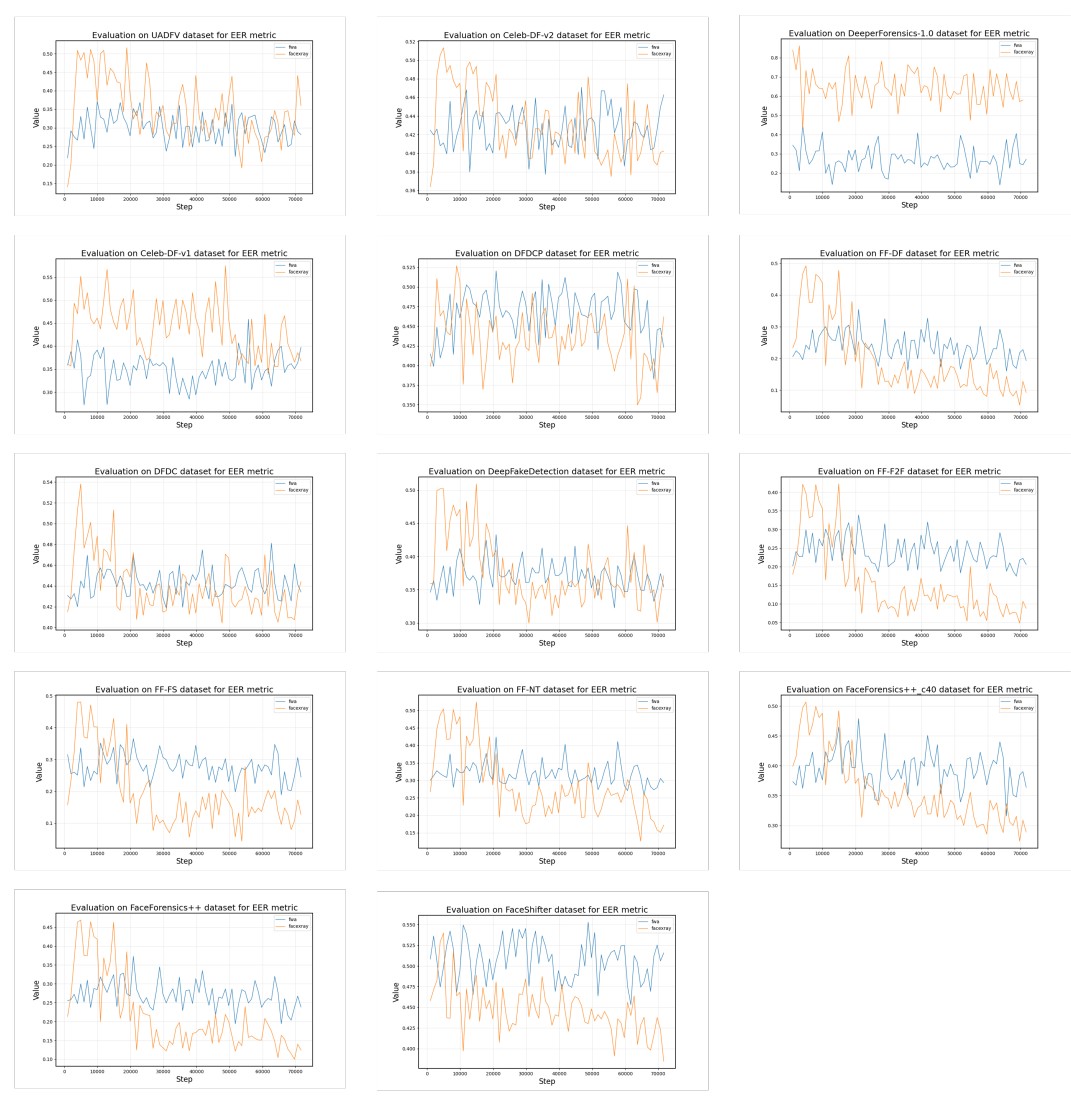

Figure 20: Illustration and visualization of all testing results during the training process. The metric is EER. We compare Face X-ray and DSP-FWA on different datasets using the EER metric.

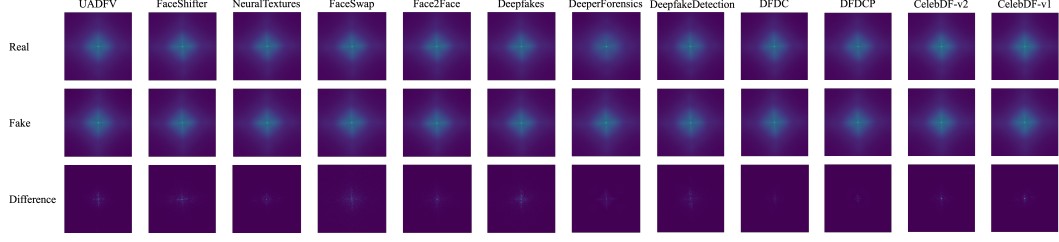

Figure 21: Frequency analysis on each dataset. We present the average spectra of high-pass filtered images, focusing on both real and fake images. Our findings align with those reported in work [48]. We observe that the shown deepfake forgeries do not display obvious artifacts in the average spectra. This underscores the similarity of our results with [48].

**Is it standard practice to use Adam optimizer for deepfake detection algorithms?** In our experiments, we want to clarify it as follows:

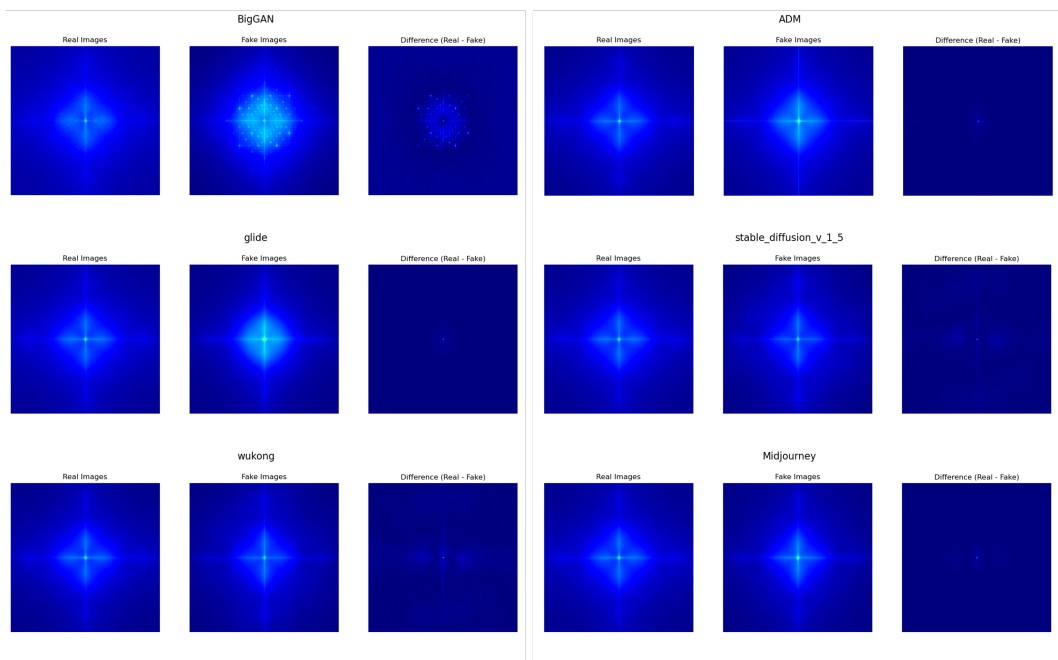

Figure 22: Frequency analysis on each dataset using the frequency analysis tool within our benchmark. We present the average spectra of high-pass filtered images, focusing on both real and fake images.

| Dataset | ResNet 34 | ResNet 50 | ResNet 152 |
|---------|-----------|-----------|------------|
| CDF-v2 | 0.7027 | 0.7491 | 0.7514 |
| DFDCP | 0.6170 | 0.6658 | 0.7078 |
| DFD | 0.6464 | 0.7002 | 0.7005 |
| FF++_c23 | 0.8493 | 0.8928 | 0.9119 |
| FF++_c40 | 0.7846 | 0.7933 | 0.8167 |
| Average | 0.7199 | 0.7602 | 0.7776 |

Table 9: *Comparing the performance of different variants of ResNet. The models are trained on FF++_c23 and tested on other datasets. The metric is the frame-level AUC.*

- Using Adam as the optimizer can be considered a common setting, and a substantial number of deepfake detection methods in existing literature have adopted this configuration (paper [20, 27, 26, 3, 12]).

- It is important to note that our benchmark is versatile and supports various optimizers, including three mainstream ones: Adam, SGD, and AdamW. In *deepfakebench*, to ensure fairness and consistency across evaluations, all detectors are trained using the Adam optimizer. Other hyperparameters, such as learning rate, are also unified as much as possible.

Overall, employing the Adam optimizer aligns with common practices in deepfake detection and serves to facilitate a consistent and equitable comparison of different algorithms.

| Detector | Condition | FF++_c23 | CDF-v2 | DFD | DFDCP | FF++_c40 | Average |
|----------|-----------|----------|--------|-----|-------|----------|---------|
| RECCE | Before Tricks | 0.9881 | 0.6083 | 0.5894 | 0.5128 | 0.5792 | 0.6556 |
| RECCE | After Tricks | 0.9621 | 0.7319 | 0.8119 | 0.7419 | 0.8190 | 0.8134 |
| RECCE | Improvement (%) | -2.63% | 20.34% | 37.84% | 44.72% | 41.37% | 24.08% |
| Xception | Before Tricks | 0.9893 | 0.5175 | 0.5870 | 0.4894 | 0.5420 | 0.6250 |
| Xception | After Tricks | 0.9637 | 0.7365 | 0.8163 | 0.7374 | 0.8261 | 0.8160 |
| Xception | Improvement (%) | -2.59% | 42.31% | 39.08% | 50.67% | 52.40% | 30.56% |
| SRM | Before Tricks | 0.9882 | 0.5993 | 0.6029 | 0.5995 | 0.5754 | 0.6731 |
| SRM | After Tricks | 0.9576 | 0.7552 | 0.8120 | 0.7408 | 0.8114 | 0.8154 |
| SRM | Improvement (%) | -3.10% | 26.00% | 34.74% | 23.56% | 40.99% | 21.13% |

Table 10: *Comparison of performance among three detectors: Naive detector (Xception), frequency detector (SRM), and spatial detector (RECCE) under the conditions of without vs. with tricks. We demonstrate that the performance of the Naive detector can be effectively enhanced through the use of tricks: data augmentation and pre-training. The models are trained on FF++_c23. The metric is the frame-level AUC.*

| Training Config | Face X-ray & FWA | Other Detectors |
|-----------------|------------------|-----------------|
| Image Size | 256, 256 | 256, 256 |
| Weight Initialization | ImageNet Pre-trained | ImageNet Pre-trained |
| Optimizer | Adam | Adam |
| Base Learning Rate | 2e-4 | 2e-4 |
| Weight Decay | 5e-4 | 5e-4 |
| Optimizer Momentum | B1, B2=0.9, 0.999 | B1, B2=0.9, 0.999 |
| Batch Size | 16 | 32 |
| Training Epochs | 10 | 10 |
| Learning Rate Schedule | None, Constant | None, Constant |
| Flip Probability | 0.5 | 0.5 |
| Rotate Probability | 0.5 | 0.5 |
| Rotate Limit | [-10, 10] | [-10, 10] |
| Blur Probability | 0.5 | 0.5 |
| Blur Limit | [3, 7] | [3, 7] |
| Brightness Probability | 0.5 | 0.5 |
| Brightness Limit | [-0.1, 0.1] | [-0.1, 0.1] |
| Contrast Limit | [-0.1, 0.1] | [-0.1, 0.1] |
| Quality Lower | 40 | 40 |
| Quality Upper | 100 | 100 |

Table 11: *The results of our main experiments (shown in Table. 2 and Figure. 2 of the manuscript) are generated using the following settings.*

