# OpenReview forum: "DeepfakeBench: A Comprehensive Benchmark of Deepfake Detection"
_NeurIPS.cc/2023/Track/Datasets_and_Benchmarks — NeurIPS 2023 Datasets and Benchmarks Poster_

### Official Review · Reviewer_oARM · 2023-07-20
**Comprehensive benchmark on deepfake detection**

**Rating:** 6
**Confidence:** 3
**Clarity:** The paper is very easy to follow.

**Strengths:**

1. The codebase implemented 15 algorithms and supports 9 datasets, which look comprehensive.
2. A systematic evaluation was performed.
3. The analysis is detailed and contains figures for intuitive visualization.

**Additional Feedback:**

Tab1. "Unpublished code provided by the authors" -> "Unpublished code, reproduced by us". The original statement is ambiguous because "authors" could mean "original paper's authors".

**Correctness:**

I have doubts on the discussion of the effect of number of training frames, namely, "all three methods tends to exhibit overfitting as the number of frames increases". Generally speaking, more data leads to less overfitting. It is more training iterations that may lead overfitting. I would suggest compare them in the setting when the total training iterations for all experiments are fixed to a same value.

**Documentation:**

/

**Ethics:**

No, there is no new dataset.

**Limitations:**

Limitations were discussed in main text.

**Opportunities For Improvement:**

1. The paper lacks a comprehensive comparison with other codebases. Firstly, the related codebases and other efforts to build deepfake detection tool-boxes (if any) should be reviewed. Secondly, a table that lists algorithm/dataset coverage of related codebases is a good addition to highlight the contribution this work.
2. Video-level evaluation could be added even with the absence of video-level detectors.

**Relation To Prior Work:**

See above comments.

**Summary And Contributions:**

This paper presents a DeepfakeBench codebase that consists of 15 deepfake detection methods and 9 datasets. Notably, four of the baseline codes are reproduced by the authors. The collection of implementations facilitates fair comparisons among these methods, and benefits future researchers on comparing baselines. The paper tested these methods in terms of within-domain and cross-domain evaluations and analyzed the effect of augmentation, backbone, choice of frames, etc., on the model performances. These results are useful for deeper understanding of relevant methods.

---

> ### Author Response · Authors · 2023-08-21
> **Response to Reviewer oARM [5]**
>
> **Q1.** ***Video-level evaluation could be added even with the absence of video-level detectors.***
>
> **R1.** Thanks for the valuable suggestion.
> * **Code accessibility:** We have updated the necessary codes to facilitate video-level evaluations on our GitHub repository, and it is accessible to all interested researchers and users.
> * **Video-based detectors implementation:** It is worth noting that while we have made the initial code available, the full-scale implementation of video-level evaluation within our framework will require meticulous development and rigorous testing. We are committed to this endeavor and excited about its potential to deepen our understanding and capabilities in this field.
>
> **Q2.** ***Generally speaking, more data leads to less overfitting. It is more training iterations that may lead to overfitting. I would suggest comparing them in the setting when the total training iterations for all experiments are fixed to the same value.***
>
> **R2.** Thanks for the valuable suggestion. Generally, we would expect that having more data would lead to less overfitting. However, we found that overfitting can occur even with an increased number of frames per video, primarily due to the following reasons:
>
> * **Data similarity:**
>     * **Hypothesis:** It is important to emphasize that the key to reducing overfitting lies in **the similarity of the data** rather than the sheer quantity. We hypothesize that many frames within a single video contain highly similar, redundant information, further leading to the overfitting problem.
>     * **Verification experiments:** To verify this idea, we conduct experiments by comparing models trained on 32 frames per video with those trained on 270 frames per video. The results, as illustrated in **Table 1 below**, show a **marked increase in overfitting when using 270 frames.** Specifically, the model trained on 270 frames quickly converged with a training loss nearing zero, while achieving an AUC close to 100% on the within-domain FF++ dataset. However, its performance on cross-domain datasets significantly deteriorated compared to the model trained on 32 frames. **The full results can be found in Figure 15 of the revised supplementary.**
>
> * **Iterations:** Our experimental results also reveal that the model trained on 270 frames underwent about **300,000 iterations**, whereas the model trained on 32 frames had only around **40,000 iterations**. Despite the large difference in iterations, the model with 270 frames converged rapidly, suggesting that **overfitting is due to data similarity, not the number of iterations**. **The full results can be found in Figure 15 of the revised supplementary.**
>
> Overall, our analysis and empirical evidence suggest that **the critical factor for reducing overfitting is data similarity, rather than simply increasing the number of frames or training iterations.**
>
> **Q3.** ***Tab1. "Unpublished code provided by the authors" -> "Unpublished code, reproduced by us". The original statement is ambiguous because "authors" could mean "original paper's authors".***
>
> **R3.** Thanks for the valuable suggestion. We have clarified them in the revised manuscript.
>
> **Table 1: Comparing the performance of Xception using the different number of training frames (270 vs. 32). We compare the within-domain training loss and testing AUC, as well as the cross-domain testing AUC (Celeb-DF-v2) at specific steps. The AUC values are presented in the format of 270 frames / 32 frames for each step.**
> | 270 / 32    | Step-5000    | Step-10000 | Step-20000 | Step-30000   | Step-40000 |
> |------------|---------|----------|------------|-------|-------|
> | Training Loss     | 0.038 / 0.178  | 0.032 / 0.146   | 0.033 / 0.119     | 0.025 / 0.105 | 0.028 / 0.107 |
> | Within-domain Testing | 0.989 / 0.941  | 0.992 / 0.947   | 0.990 / 0.958     | 0.992 / 0.961 | 0.992 / 0.963 |
> | Cross-domain Testing | 0.596 / 0.699  | 0.620 / 0.690   | 0.614 / 0.703     | 0.621 / 0.722 | 0.575 / 0.656|

---

> ### Author Response · Authors · 2023-08-28
>
> Dear Reviewer oARM:
>
> We would like to express our sincere gratitude for your valuable insights and suggestions on our work.
>
> We have tried our best to address the concerns during the rebuttal process. However, we would greatly appreciate knowing whether our response has effectively resolved your doubts. Your feedback will be instrumental in improving the quality of our work. **As the end of the discussion period is approaching, we eagerly await your reply before the end.**
>
> Sincerely,
>
> Authors

---

> > ### Comment · Reviewer_oARM · 2023-08-29
> > **Response to rebuttal**
> >
> > Thanks for the rebuttal. I still cannot get the explanation on overfitting.
> >
> > > Hypothesis: It is important to emphasize that the key to reducing overfitting lies in the similarity of the data rather than the sheer quantity. We hypothesize that many frames within a single video contain highly similar, redundant information, further leading to the overfitting problem.
> >
> > I cannot get why similar frames causes overfitting. Suppose a video has 320 frames, and only 32 frames are unique, and each unique frame is repeated for 10 times. Granted, it is highly similar and redundant. But what is the difference between using all 320 frames and using only 32 unique frames? Shouldn't the result be same, considering they are trained using exactly the same data?
> >
> > A plausible explanation in my head is that the training batch size is too small (32). If a video provides more frames, then it is more likely that frames from a single video may dominate a batch and then hurt the training.
> >
> > I am not convinced by the argument "more data leads to overfitting", even if the added data are repetitive.

---

> > > ### Author Response · Authors · 2023-08-29
> > >
> > > We would like to express our sincere gratitude to the reviewer for pointing out this essential and insightful concern. Upon initial reflection, it is indeed logical to assume that using 320 repetitive frames should yield the same result as using 32 unique frames.
> > >
> > > However, our seemingly counterintuitive findings are actually in line with, and can be explained by, recent research in the domain.
> > > * Prior research [1] has shown that **deepfake detectors might be unintentionally overfitted to features that are not directly related to forgery, consequently undermining the model's ability to generalize.** This implies that not all information contained in an image is beneficial for deepfake detection; some elements may actually impair the model's ability to generalize.
> > > * Another study [2] further elaborates on this by demonstrating that **deepfake detectors can inadvertently learn the identity (ID) features of a face**, which can **detract from their ability to focus on the actual forgery-related features** (as discussed in Section 3.1 of [2]).
> > >
> > > Therefore, in a scenario where we have 320 frames, there is a possibility that **the model could mistakenly memorize the ID or other forgery-unrelated features of the image, leading to poor generalization as it might not be able to effectively learn the features related to forgery.** Furthermore, we have implemented a shuffle operation during data reading to ensure that the images in a batch predominantly originate from different videos, thereby eliminating the possibility of a single video dominating a batch.
> > >
> > > We hope this response can address the concern raised. Thank you for your invaluable suggestions and feedback; they are incredibly helpful!
> > >
> > > References:\
> > > [1] Exploring Disentangled Content Information for Face Forgery Detection. ECCV 2022.\
> > > [2] Implicit Identity Leakage: The Stumbling Block to Improving Deepfake Detection Generalization. CVPR 2023.

---

> > > > ### Comment · Reviewer_oARM · 2023-08-29
> > > > **Response to comment**
> > > >
> > > > Thanks for the explanation. The "deepfake detectors can inadvertently learn the identity (ID) features of a face" argument makes sense to me. I have no further question.

---

> > > > > ### Author Response · Authors · 2023-08-29
> > > > >
> > > > > We are glad to hear that the explanation clarified your concerns. Thank you once again for your insightful questions and feedback!

---

### Official Review · Reviewer_ZyZY · 2023-07-20
**Overall a fairly comprehensive benchmark**

**Rating:** 6
**Confidence:** 3
**Clarity:** The paper is generally well-written a…

**Strengths:**

- Overall the submission is fairly solid. The paper is generally well-written and well-motivated. It would be a good contribution to the community.
- The benchmark has considerable size (9 datasets and 15 baseline methods).
- The authors considered a fairly comprehensive set of evaluation techniques and metrics.
- The work also goes further and provide useful empirical analyses regarding implementation details such as the effect of the number of training frames and data augmentation.
- The provided codebase seems to have a good degree of completeness and is generally well-documented.

**Additional Feedback:**

- L190: Is it standard practice to use Adam optimizer for deepfake detection algorithms?

**Correctness:**

Overall the paper looks correct on a technical level. The presented datasets and the metrics for evaluating on the datasets look reasonable. The experiment setup is also reasonable.

**Documentation:**

Overall the documentation is satisfactory. URL to code repo is included, though maintenance plan is not discussed.

**Ethics:**

Since the benchmark involves human facial data, it may be helpful to include more discussions or review for consent, privacy, responsible use, and legal compliance.

**Limitations:**

The authors have provided discussions on how the proposed benchmark may facilitate the development of anti-deepfake methods. It would be appreciated if negative societal impact are also dicussed; e.g. how the collected datasets may be misused.

**Opportunities For Improvement:**

- Presentation: In general, the figures and tables in the main text are a bit too small (Table 2, Figure 2-4, etc.).
- Qualitative analyses across datasets / baseline methods: While quantitative metrics should be the primary form of evaluation, it may also help to include more visualization results on why certain methods perform better on certain datasets; what are the failure cases; etc.
- Details of hyperparameters: While there is a codebase accompanies the submission, the details on hyperparameters are not entirely clear from the main text or the supplementary. It would be ideal to have a table outlining all the relevant hyperparameter setting used to produce the experimental results (e.g. table 2).

**Relation To Prior Work:**

The paper provided good literature review and discusses key connections/differences to prior work.

**Summary And Contributions:**

- The paper proposes DeepfakeBench, a benchmark for detecting learning-based facial image manipulation.
- The benchmark including 9 datasets, a codebase for loading/preprocessing data and training/evaluation different methods, and the implementation of 15 baseline methods.
- Contributions include
    - standardizing (formatting, preprocessing, …) different datasets under a single codebase so that they are readily accessible
    - providing standard implementations of baseline methods, which may otherwise be using different configurations, making experimental results difficult to compare
    - providing a codebase for standardizing the training and evaluation of different methods, which may otherwise be using different metrics and evaluating on frame-level vs video-level
    - providing new empirical insights using the provided evaluation framework and codebase

---

> ### Author Response · Authors · 2023-08-21
> **Response to Reviewer ZyZY [4]**
>
> **Q1.** ***Details of hyperparameters: While there is a codebase accompanies the submission, the details on hyperparameters are not entirely clear from the main text or the supplementary. It would be ideal to have a table outlining all the relevant hyperparameter setting used to produce the experimental results (e.g., table 2).***
>
> **R1.** Thanks. The results of our main experiments (shown in **Table. 2 and Figure. 2 of the manuscript**) are generated using the following settings:
> | Training config         | Face X-ray & FWA                | Other Detectors               |
> |-------------------------|----------------------------------|----------------------------------|
> | Image Size              | 256, 256                         | 256, 256                         |
> | Weight Initialization   | ImageNet Pre-trained             | ImageNet Pre-trained             |
> | Optimizer               | Adam                             | Adam                             |
> | Base Learning Rate      | 2e-4                             | 2e-4                             |
> | Weight Decay            | 5e-4                             | 5e-4                             |
> | Optimizer Momentum      | B1, B2=0.9, 0.999                | B1, B2=0.9, 0.999                |
> | Batch Size              | 16                               | 32                               |
> | Training Epochs         | 10                               | 10                               |
> | Learning Rate Schedule  | None, Constant                    | None, Constant                    |
> | Flip Probability        | 0.5                              | 0.5                              |
> | Rotate Probability      | 0.5                              | 0.5                              |
> | Rotate Limit            | [-10, 10]                        | [-10, 10]                        |
> | Blur Probability        | 0.5                              | 0.5                              |
> | Blur Limit              | [3, 7]                           | [3, 7]                           |
> | Brightness Probability  | 0.5                              | 0.5                              |
> | Brightness Limit        | [-0.1, 0.1]                      | [-0.1, 0.1]                      |
> | Contrast Limit          | [-0.1, 0.1]                      | [-0.1, 0.1]                      |
> | Quality Lower           | 40                               | 40                               |
> | Quality Upper           | 100                              | 100                              |
>
> An important observation is that the batch size used for FWA and Face X-ray differs by being half that of other detectors. This discrepancy arises from the fact that the input data for these detectors is in the form of pairs. We have provided explicit clarification regarding this particular configuration in **lines 304-306 of the original supplementary material**.
>
>
> **Q2.** ***It would be appreciated if negative societal impact is also discussed, e.g., how the collected datasets may be misused.***
>
> **R2.** We appreciate the reviewer's emphasis on the societal impacts associated with our work. While our tool is designed to have positive contributions in the field of deepfake detection, we also recognize the potential risks associated with the misuse of the collected datasets.
> * Regarding ethical and responsible use of our datasets, we are contemplating measures to mitigate such risks.
> * As a concrete step towards this goal, we are planning to **implement controlled access mechanisms and usage agreements akin to the practices observed in the ImageNet dataset**. These measures are intended to promote the responsible and ethical use of our datasets and to mitigate the possibility of negative societal consequences stemming from their misuse.
>
> **Q3.** ***Is it standard practice to use Adam optimizer for deepfake detection algorithms?***
>
> **R3.** Thanks. We want to clarify it as follows:
> * Using Adam as the optimizer can be considered a common setting, and a substantial number of detectors have adopted this configuration (paper [3,4,5,6]).
> * It is important to note that our benchmark is versatile and supports various optimizers, including three mainstream ones: Adam, SGD, and AdamW. In deepfakebench, to ensure fairness and consistency across evaluations, all detectors are trained using the Adam optimizer.
>
> Overall, employing the Adam optimizer aligns with common practices and facilitates a fair comparison.
>
> **Q4.** ***Figures and tables in the main text are a bit too small.***
>
> **R4.** Thanks. We have clarified them in the revised manuscript.
>
> [3] Face X-ray for More General Face Forgery Detection. CVPR 2020.\
> [4] Generalizing Face Forgery Detection with High-frequency Features. CVPR 2021.\
> [5] Self-supervised Learning of Adversarial Example: Towards Good Generalizations for Deepfake Detection. CVPR 2022.\
> [6] Implicit Identity Leakage: The Stumbling Block to Improving Deepfake Detection Generalization. CVPR 2023.\

---

> > ### Comment · Reviewer_ZyZY · 2023-08-28
> > **Thanks for the response; my concerns are mostly resolved**
> >
> > I appreciate the authors for providing a response. My concerns are mostly resolved. It would be great to include the above details in the updated version.

---

> > > ### Author Response · Authors · 2023-08-28
> > > **Thanks for your feedback; we have updated the above details in the revised supplementary.**
> > >
> > > We greatly appreciate your feedback and are strongly encouraged that our rebuttals have well addressed your concerns. We have made the necessary updates in the revised supplementary material and manuscript as follows:
> > >
> > > 1. **Details of hyperparameters:** The details of the hyperparameters can be found in **Table 10 of the revised supplementary material.**
> > > 2. **Discussion of the negative societal impact:** We have included a discussion on the potential negative societal impacts in **Lines 319-324 of the revised manuscript.**
> > > 3. **Discussion of optimizer:** Further discussion on the Adam optimizer can be found in **Lines 520-530 of the revised supplementary material.**
> > > 4. **Font size adjustment:** The font size in **Table 2 of the revised manuscript** has been increased for better readability.

---

### Official Review · Reviewer_fUyn · 2023-07-21
**This paper introduces DeepfakeBench, the comprehensive benchmark for deepfake detection, and this paper has a clear logic and well-organized narrative.**

**Rating:** 6
**Confidence:** 5

**Strengths:**

This paper provides a unified, standardized, and comprehensive benchmark for deepfake detection, reducing unfair comparisons arising from different experimental setups, datasets, and data processing methods. It facilitates further in-depth research in this field and offers more comprehensive evaluation metrics.

**Additional Feedback:**

N/A

**Clarity:**

The paper categorizes the detection methods into three groups, but there are two inconsistent descriptions between the first page “spatial-based detector, and frequency-based detector” and the second page “spatial detector, and frequency detector”. Additionally, for the names of the three detectors in Figure 1, it is recommended to ensure consistency. In addition, I think that this paper has a clear logic and a well-organized narrative.

**Correctness:**

This paper provides a unified, standardized, and comprehensive benchmark for deepfake detection, consisting of three core modules: data processing module, training module, and evaluation and analysis module. The design of the workflow is reasonable, and the results are thoroughly, conveniently, and comprehensively evaluated.

**Documentation:**

The paper provides the original code and usage instructions for the proposed benchmark, along with a user-friendly interface that allows users to modify it according to their specific application scenarios. In the supplementary material, additional explanations about the experimental details are provided, facilitating the reproducibility of the entire framework and its broader utilization.

**Ethics:**

I believe that the submitted files do not present any ethical issues and do not require further discussion or review.

**Limitations:**

This paper analyzes the impact of data augmentation methods, backbone networks, pre-training techniques, and other factors on various methods. The experimental results reveal some noteworthy "counterintuitive" phenomena. The paper only describes these observations, hoping to stimulate further in-depth research in these areas. Additionally, in this paper, DeepfakeBench primarily focuses on providing algorithms and evaluations at the frame level, and it is expected that future work will extend the analysis, detection, and evaluation to the video level, thus forming a more comprehensive framework.

**Opportunities For Improvement:**

(1)The paper presents detailed information on 15 published methods in Table 1, including whether their code has been made open-source. It can be observed that some methods have not released their code publicly. However, in Section 3.1, it is mentioned that the selection of detectors is based on the third criterion, which emphasizes the ease of implementation. Therefore, non-open-source methods were not included, indicating a discrepancy in the description between these sections.
(2)The 15 methods listed in Table 1 have a relatively small proportion of recent approaches. It is recommended to include some of the latest relevant methods from the years 2022 and 2023 to supplement the existing ones.
(3)In Section 4.2, a cross-domain evaluation is designed, and the results are presented in Table 2. However, the direction of the cross-domain evaluation experiment is not specified and requires clarification.
(4) In the visualization of different data augmentation methods in Figure 4, the results are sorted using different colors instead of representing each method with a single color. This causes the vertical axis coordinates to change continuously, leading to the unclear presentation of the results. It is recommended to adjust the display format of the results to improve readability by using a more consistent and clear representation method, such as using one color for each method.

**Relation To Prior Work:**

The paper introduces previous relevant Benchmarks and datasets, but they focus on their own datasets and do not provide a standardized approach for handling data from different datasets. This may lead to inconsistencies and hinder fair comparisons. Additionally, the lack of a unified framework in some benchmarks can result in variations in training strategies, settings, and augmentation methods, potentially leading to differences in the obtained results.

**Summary And Contributions:**

This paper introduces DeepfakeBench, the comprehensive benchmark for deepfake detection, comprising three fundamental contributions: 1) a unified data management system ensuring consistent input across all detectors, 2) an integrated framework for implementing 15 state-of-the-art methods, and 3) standardized a series of evaluation metrics and protocols fostering transparency and reproducibility.

---

> ### Author Response · Authors · 2023-08-21
> **Response to Reviewer fUyn [3]**
>
> **Q1.** ***In Section 4.2, a cross-domain evaluation is designed, and the results are presented in Table 2. However, the direction of the cross-domain evaluation experiment is not specified and requires clarification.***
>
> **R1.** Thanks for the kind mention. We conduct the within-domain and cross-domain evaluations, as shown in **Table 2 of the manuscript**. **These detectors are all trained on the FF++ (c23) and evaluated on other datasets**, e.g., FF++ (c40), Celeb-DF-v1, etc. We have clarified it in **lines 204-207 of the original manuscript**. To further enhance clarity, we have also provided additional clarification within **the caption of Table 2 in the revised manuscript.**
>
> **Q2.** ***In this paper, DeepfakeBench primarily focuses on providing algorithms and evaluations at the frame level, and it is expected that future work will extend the analysis, detection, and evaluation to the video level, thus forming a more comprehensive framework.***
>
> **R2.** Thanks for the valuable suggestion. Currently, DeepfakeBench is primarily focused on algorithms and assessments at the frame level. However, we are actively planning to broaden our scope by extending our methodologies and evaluations to the video level (as indicated in **lines 311-313 of the original manuscript**). This expansion entails the inclusion of classical and cutting-edge temporal-based detectors within our existing benchmark framework.
> *    Notably, this endeavor will necessitate alterations to the codebase, including data IO, training architecture, evaluation protocols, and analytical procedures. Consequently, a redesign of the codebase is imperative to enhance its robustness. While a redesign is necessary, due to its significance, we will proceed with the required codebase modifications.
>
> **Q3.** ***The paper presents detailed information on 15 published methods in Table 1. It can be observed that some methods have not released their code publicly. However, in Section 3.1, it is mentioned that the selection of detectors is based on the criterion, which emphasizes the ease of implementation. Therefore, non-open-source methods were not included, indicating a discrepancy in the description between these sections.***
>
> **R3.** Thanks for your thoughtful consideration. Indeed, the criterion of selecting detectors hinges primarily on the criterion outlined in **Lines 131-143 of the original manuscript.** However, it is important to note that the **non-open-source methods we employed are derived from the code directly provided by their respective authors to ensure accuracy and consistency.** We have clarified it in **Lines 144-145 of the updated manuscript.**
>
> **Q4.** ***The 15 methods listed in Table 1 have a relatively small proportion of recent approaches. It is recommended to include some of the latest relevant methods from the years 2022 and 2023 to supplement the existing ones.***
>
> **R4.** Thanks for the valuable suggestion. We have updated two new detectors in the year 2022 to our current codebase: EDCI (Liang et al., ECCV'2022) and RFM (Wang et al., CVPR'2022). We will constantly update our benchmark by implementing more latest relevant detectors in the future.
>
> **Q5.** ***In the visualization of different data augmentation methods in Figure 3, the results are sorted using different colors instead of representing each method with a single color. Also, there are two inconsistent descriptions between the first page “spatial-based detector, and frequency-based detector” and the second page “spatial detector, and frequency detector”.***
>
> **R5.** Thanks for the valuable suggestion. We have clarified them in **the updated manuscript.**

---

> > ### Comment · Reviewer_fUyn · 2023-08-30
> > **Thanks for the responses.**
> >
> > The authors have made some improvements in this version, and I think the current version is more readable.

---

> > > ### Author Response · Authors · 2023-08-30
> > >
> > > We greatly appreciate your feedback and are pleased to know that the revisions have improved the readability of the work. Thank you once again for your valuable insights!

---

> ### Author Response · Authors · 2023-08-28
>
> Dear Reviewer fUyn:
>
> We would like to express our sincere gratitude for your valuable insights and suggestions on our work.
>
> We have tried our best to address the concerns during the rebuttal process. However, we would greatly appreciate knowing whether our response has effectively resolved your doubts. Your feedback will be instrumental in improving the quality of our work. **As the end of the discussion period is approaching, we eagerly await your reply before the end.**
>
> Sincerely,
>
> Authors

---

### Official Review · Reviewer_uRdm · 2023-07-21
**A Comprehensive Benchmark of Deepfake Detection**

**Rating:** 5
**Confidence:** 3
**Correctness:** Fine
**Clarity:** Fine

**Strengths:**

Pros:
1. This paper proposes a comprehensive benchmark on Deepfake Detection
2. This paper proposes a useful toolkit for deepfake detection.
3. This paper conduct solid experiments.

**Additional Feedback:**

No

**Documentation:**

Fine

**Ethics:**

Fine

**Limitations:**

Fine

**Opportunities For Improvement:**

Cons:
1. Relate works are incomplete. The authors only mention two related benchmarks (Line 90-93). Actually, there are many related benchmarks the authors have missed such as [1,2]. It is desired the authors conduct an explicit comparison (for example, using a comparison table) with all the related benchmarks to show the differences and remaining challenges. Otherwise, it is hard to know the unique contributions of this paper.
2. Limited insights are provided. It is desired the authors could provide more insights on detecting diffusion model-based deepfake and GAN-based deepfake.
3. Not new datasets are provided. This paper only collects the existing datasets (Line 106). No new datasets are provided.


[1] Towards Benchmarking and Evaluating Deepfake Detection
https://arxiv.org/abs/2203.02115

[2] A Continual Deepfake Detection Benchmark: Dataset, Methods, and Essentials
https://openaccess.thecvf.com/content/WACV2023/papers/Li_A_Continual_Deepfake_Detection_Benchmark_Dataset_Methods_and_Essentials_WACV_2023_paper.pdf

**Relation To Prior Work:**

Relate works are incomplete. The authors only mention two related benchmarks (Line 90-93). Actually, there are many related benchmarks the authors have missed such as [1,2]. It is desired the authors conduct an explicit comparison (for example, using a comparison table) with all the related benchmarks to show the differences and remaining challenges. Otherwise, it is hard to know the unique contributions of this paper.


[1] Towards Benchmarking and Evaluating Deepfake Detection
https://arxiv.org/abs/2203.02115

[2] A Continual Deepfake Detection Benchmark: Dataset, Methods, and Essentials
https://openaccess.thecvf.com/content/WACV2023/papers/Li_A_Continual_Deepfake_Detection_Benchmark_Dataset_Methods_and_Essentials_WACV_2023_paper.pdf

**Summary And Contributions:**

A Comprehensive Benchmark of Deepfake Detection

---

> ### Author Response · Authors · 2023-08-21
> **Response to Reviewer uRdm [2]**
>
> **Q1.** ***It is desired the authors could provide more insights on detecting diffusion model-based deepfake and GAN-based deepfake.***
>
> **R1.** We appreciate the reviewer's interest in gaining deeper insights into the detection of diffusion model-based deepfake and GAN-based deepfake. Firstly, we would like to clarify that our benchmark primarily focuses on detecting face forgery techniques, particularly those involving face swapping and face reenactment. Following your constructive suggestions, we have extended our benchmark to the detection of **not only face forgeries but also GAN-generated and diffusion-generated data**, as shown in **Lines 448-454 of our updated supplementary.**
> * **Detection Performance**: Our preliminary results (see Table 1) indicate promising detection capabilities across various scenarios:
>     * **When training and testing with the same type of data** (ADM, VQDM, SD5, wukong), the detection performance is remarkably high (AUC values close to or equal to 1.000).
>     * **When testing across different data types** (ADM, VQDM, SD5, wukong), the model maintains strong discriminatory power, with AUC values ranging from 0.8299 to 0.9999 and AP values exhibiting competitive performance.
> * **Insights into Frequency Analysis**: To delve deeper, **we employ the frequency analysis tool within our benchmark** to visualize the mean Discrete Fourier Transform (DFT) spectrum of both real and generated images.
>     * This analysis has unearthed intriguing observations specific to diffusion-generated images when contrasted with GAN-generated images.
>     * Particularly, **diffusion-generated images exhibit fewer artifacts, while GAN-generated images display a noticeable checkerboard pattern of artifacts**. Noteworthy differences in artifacts are also noted with specific algorithms, such as "wukong" and "stable-diffusion-v5" (see **Figure 17 of the revised supplementary**).
>
> **Table 1: Preliminary results obtained by deploying the Xception model on diffusion-generated data using our DeepfakeBench codebase.**
> | Training Data | Test Data | Resolution |      AUC |       AP |
> |---------------|-----------|------------|----------|----------|
> | ADM           | ADM       | 256        |    1.000 |    1.000 |
> | ADM           | VQDM      | 256        |    0.8299|    0.7645|
> | VQDM          | ADM       | 256        |    0.8538|    0.8488|
> | VQDM          | VQDM      | 256        |    0.9993|    0.9992|
> | SD5           | SD5       | 512        |    0.9999|    0.9999|
> | SD5           | SD4       | 512        |    0.9998|    0.9996|
> | SD5           | wukong    | 512        |    0.9999|    0.9997|
> | wukong        | SD5       | 512        |    0.9995|    0.9997|
> | wukong        | SD4       | 512        |    0.9998|    0.9996|
> | wukong        | wukong    | 512        |    0.9999|    0.9999|
>
> **Q2.** ***Not new datasets are provided. This paper only collects the existing datasets (Line 106). No new datasets are provided.***
>
> **R2.** We genuinely appreciate the suggestion.
> * **First**, indeed, our current emphasis is not centered around introducing new datasets. Instead, our benchmark's core value at this juncture is to construct a unified, modular, and user-friendly codebase that facilitates consistent inputs and standardized evaluations across various detection methods. In addition, another consideration is that generating a new deepfake dataset may involve several issues, such as possible negative social impacts, copyright, privacy, etc.
> * **Second**, as we move forward, we acknowledge the significance of incorporating additional datasets to further assess detector performance comprehensively. While the current datasets we have collected are very comprehensive, we will keep eye on new deepfake datasets in the field, and include them to make the evaluations in our benchmark more comprehensive.

---

> ### Author Response · Authors · 2023-08-28
>
> Dear Reviewer uRdm:
>
> We would like to express our sincere gratitude for your valuable insights and suggestions on our work.
>
> We have tried our best to address the concerns during the rebuttal process. However, we would greatly appreciate knowing whether our response has effectively resolved your doubts. Your feedback will be instrumental in improving the quality of our work. **As the end of the discussion period is approaching, we eagerly await your reply before the end.**
>
> Sincerely,
>
> Authors

---

> > ### Comment · Reviewer_uRdm · 2023-08-28
> > **Thanks for the response.**
> >
> > Thanks for the response. It is suggested the authors highlight the changes in different colors. Otherwise, it is hard to see what part the authors have revised. The authors may consider providing more insights on detecting diffusion model-based deepfake and GAN-based deepfake in the main paper rather than the supplementary. I will raise the score considering the efforts.

---

> > > ### Author Response · Authors · 2023-08-29
> > >
> > > Dear Reviewer uRdm:
> > >
> > > Thank you for your valuable feedback and understanding of the efforts we have made to improve our work.
> > >
> > > * **More insights on detecting diffusion-based deepfake in the main paper:** We truly appreciate your thoughtful suggestion. However, please understand that **incorporating this content directly into the main paper at this stage would unfortunately lead to a renumbering of the lines.** This, in turn, could create confusion during the review process as the line numbers referred to in previous reviews and our responses would no longer match. We hope you understand this logistical challenge. Please be assured that this important content will be diligently integrated into the appropriate sections of the final version of the main paper.
> > >      * In the analysis section of the final main paper, **we will present the detection results obtained using our benchmark and delve into the analysis of various artifacts using the frequency analysis tools included in our benchmark.** This comprehensive analysis will provide valuable insights into the characteristics and detection of these types of deepfakes.
> > > * **Highlight the changes in different colors:** Following your suggestion, we have highlighted the changes in the revised version of the manuscript and supplementary material using red color.
> > >
> > > Once again, thank you for your constructive feedback, and we hope that our response addresses your concerns satisfactorily. We appreciate your consideration in raising the score and look forward to your final feedback.

---

### Official Review · Reviewer_Hoqi · 2023-07-23

**Rating:** 7
**Confidence:** 3
**Correctness:** The evaluation methods and experiment…
**Clarity:** The paper is well written.

**Strengths:**

1. The paper provides a unified platform for comparing deepfake detection methods, which would facilitate the evaluation of future detection methods to be developed.

2. The codebase has a modular design and looks easy to use.

3. The evaluation is comprehensive.

**Additional Feedback:**

None.

**Documentation:**

There is sufficient detail to support reproducibility.

**Ethics:**

No.

**Limitations:**

The authors have adequately addressed the limitations.

**Opportunities For Improvement:**

The paper was well-written and does not have any prominent weaknesses.

One opportunity for improvement is that part of the experiment section only summarized the phenomena observed in the experiments, yet lacking an explanation of why they happened. For example, why do Naive detectors work perform as well as more advanced in certain settings? Also, when analyzing the effect of backbone architecture, the paper mentioned that Xception and EfficientNet-B4 work better than ResNet 34, but didn't give an in-depth explanation of why the former two architectures are more advantageous.


**Relation To Prior Work:**

It is clearly discussed how the paper differs from previous contributions.

**Summary And Contributions:**

The paper introduces a comprehensive benchmark for deepfake detection. The contributions include a modular-based codebase, thorough evaluation and analysis of existing algorithms.

---

> ### Author Response · Authors · 2023-08-21
> **Response to Reviewer Hoqi [1]**
>
> **Q1.** ***Why do Naive detectors work perform as well as more advanced in certain settings?***
>
> **R1.** Thanks for this insightful concern. In our experiments, we have observed that Naive detectors can exhibit competitive performance compared to more complex methods, which might be surprising given the advancements in the field. We would like to explain this phenomenon from the following aspects.
>
> * **Consistency in experimental procedures and evaluation metrics:** It is worth noting that comparison methodologies can vary across studies, and **directly adopting results from prior papers could sometimes lead to discrepancies.** For instance, current studies often directly cite the results of Xception from the original paper [1], but **differences in factors like training frame quantity used in different works can result in disparities.** Further investigation can be seen in the **Line 415-429 in the revised supplementary.**
> * **Why Naive detectors in our benchmark work?**
>     * Naive detectors, **despite their simplicity, may have inherent strengths that are yet to be fully understood and harnessed.** However, few previous studies have deeply identified the conditions under which they can be particularly effective.
>     *  Previous works have shown that some tricks could bolster the performance of Naive detectors, **e.g., pre-training or data augmentation**. To this end, we conduct an experiment to compare the performance between the Naive detector and the complex under the conditions with or without tricks. **By adding the tricks, we find the gap between the Naive detector and complex detector is reduced** (see **Table 1 below**). Showing that the performance of the Naive detector is effectively mined by using these tricks.
> We have clarified it to the **Line 499-519 of our updated supplementary.**
>
> **Q2.** ***When analyzing the effect of backbone architecture, the paper mentioned that Xception and EfficientNet-B4 work better than ResNet 34, but why the former two architectures are more advantageous?***
>
> **R2.** Thanks. After our preliminary investigation, we find that the reasons are related to two factors, namely **architecture and models' scale**:
> * **First**, we identify a **common module** in EfficientNet and Xception that is not present in ResNet, namely **the depthwise separable convolution module**. We hypothesize that this module might be contributing to the performance advantage. To evaluate this, we insert this module into ResNet, replacing only the first convolutional layer. Experiments demonstrate significant improvements on many test datasets (see **Table 2 below**).
> * **Second**, additional factors that might also exert an impact on the ultimate performance come to light. These encompass the **number of layers** within the architecture and **the number of parameters** associated with it.
>     * **Number of Parameters**: Referring to ****Table 6 in the revised supplementary****, it becomes evident that the parameter numbers remain comparable among the three models.
>     * **Number of layers**: Subsequently, a comprehensive exploration is conducted to assess the impact of layer numbers. This assessment involves a diverse range of ResNet variants, including ResNet 50 and ResNet 152. Results in ****Table 7 in the revised supplementary**** uncover that ResNet 50, characterized by a greater number of layers in comparison to ResNet 34, yields a substantial enhancement in performance.
>
> The above analysis has been added in **the updated supplementary, Line 480-498**.
>
> **Table 1: Comparison of performance among Naive detector (Xception), frequency detector (SRM), and spatial detector (RECCE) under the conditions of w vs. wo tricks. The tricks we apply are data augmentation, pre-training, and fewer training frames (from 270 to 32 frames of each video).**
> | Detector |  FF++_c23 | CDF-v2 | DFD   | DFDCP | FF++_c40 |
> |----------|----------|--------|-------|-------|----------|
> | Xception (wo tricks)    | 0.989    | 0.517  | 0.587 | 0.489 | 0.542    |
> | Xception (w tricks)   | 0.964    | 0.737  | 0.816 | 0.737 | 0.826    |
> | RECCE (wo tricks)     | 0.988    | 0.608  | 0.589 | 0.512 | 0.579    |
> | RECCE  (w tricks)    | 0.962    | 0.732  | 0.812 | 0.742 | 0.819    |
> | SRM    (wo tricks)     | 0.988    | 0.599  | 0.602 | 0.599 | 0.575    |
> | SRM   (w tricks)      | 0.958    | 0.755  | 0.812 | 0.741 | 0.811    |
>
> **Table 2: Ablation study regarding the effectiveness of the depthwise separable convolution module for ResNet.**
> | Model      | FF++_c23    | FF++_c40 | CDF-v2 | DFD   | DFDCP | UADFV  | Average |
> |------------|---------|----------|------------|-------|-------|--------|---------|
> | ResNet     | 0.8493  | 0.7846   | 0.7027     | 0.6464| 0.6170| 0.8739 | 0.7456  |
> | ResNet-dsc | 0.8968  | 0.8048   | 0.7582     | 0.7006| 0.6766| 0.8895 | 0.7877  |
> | Improvement| +5.60%  | +2.57%   | +7.90%     | +8.39%| +9.64%| +1.78% | +5.64%  |
>
> [1] FaceForensics++. ICCV 2019.

---

> ### Author Response · Authors · 2023-08-28
>
> Dear Reviewer Hoqi:
>
> We would like to express our sincere gratitude for your valuable insights and suggestions on our work.
>
> We have tried our best to address the concerns during the rebuttal process. However, we would greatly appreciate knowing whether our response has effectively resolved your doubts. Your feedback will be instrumental in improving the quality of our work. **As the end of the discussion period is approaching, we eagerly await your reply before the end.**
>
> Sincerely,
>
> Authors

---

### Author Response · Authors · 2023-08-21
**Common Response**

**Q1.** (**From Reviewer uRdm, oARM**) ***It is desired to conduct an explicit comparison with all the related benchmarks to show the differences and remaining challenges.***

**R1.** Thanks for the valuable suggestion. Reviewer uRdm identifies two related benchmarks (paper [1,2]). Let's discuss and compare our benchmark with these two works:
* **Compare with paper [1]**: Paper [1] generally aligns with the focus of our benchmark. To facilitate a concise comparison, we have created **Table 9 of the revised supplementary** to **highlight 11 key aspects**. This table clearly emphasizes our benchmark's distinct contributions and differentiates it from paper [1].
* **Compare with paper [2]**: There are distinct differences in focus between paper [2] and our work, which can be outlined as follows:
    * **Different scopes of deepfake:** It is crucial to note that **our focus and the scope of paper [2] may differ to a considerable extent**. While paper [2] focuses solely on detecting GAN-generated images, our work takes a broader manipulation approach by providing evaluations for detecting both GAN and diffusion-generated data, as well as AI face-swapping forgeries.
    * **Different detection methods:** **The varying scope of deepfake types necessitates different methods for detecting forgeries.** Specifically, GANs and face-swapping, may leave unique artifacts, necessitating varied detection methods. As a result, our work and paper [2] utilize distinct algorithms and datasets, making direct comparisons challenging.
    * **Other differences:** The primary objective of our benchmark revolves around furnishing a **comprehensive and modular codebase for data preprocessing, training, and a series of analysis tools.** In contrast, paper [2] introduces a benchmark **specifically tailored for detecting GAN-generated images using continual learning.**

Following the suggestion, we have added the citations and comparisons with the above two works in our updated manuscript (In **Sec. 2, Lines 100-104**).

[1] Towards Benchmarking and Evaluating Deepfake Detection. ArXiv 2022.
[2] A Continual Deepfake Detection Benchmark: Dataset, Methods, and Essentials. WACV 2023.

**Q2.** **(From Hoqi, uRdm, fUyn, ZyZY)** ***More explanations are needed to further explore the phenomena observed in the experiments to show more insights.***

**R2.** We sincerely appreciate the insightful suggestions. In our benchmark, we have conducted comprehensive and fair evaluations for the **cross-data evaluation** (see **Table 1 in the manuscript**) and **cross-manipulation evaluation** (see **Figure 2 in the manuscript**). Here, we present an in-depth analysis of two intriguing observations from the above two evaluations, as follows.
1. ***Cross-Manipulation Evaluation* and the Efficacy of Blending-Based Detectors:**
    * **Observation:** As shown in **Figure 2 of the manuscript**, blending-based detectors, specifically Face X-ray and FWA, exhibit superior performance in cross-manipulation evaluation. We have discussed these findings in lines 227-230 of the manuscript.
    * **Analysis:** To delve deeper into the reasons behind this efficacy, we conduct a t-SNE analysis (see **Figure 18 of the updated supplementary**), visualizing labels in the feature space. Our findings suggest that images generated through blending technology exhibit distinctiveness, distancing them from images generated by alternative manipulation methodologies. **This characteristic enlarges the forgery space, culminating in enhanced generalization capabilities.**
2. ***Cross-Data Evaluation* and the Importance of Phase Spectrum:**
    *  **Observation:** In **Table 2 of the manuscript**, we highlight the SPSL detector, which achieves an impressive average score of 78.75% in cross-domain evaluation. **A distinctive feature of SPSL compared to Xception is the incorporation of the phase spectrum.**
    *  **Analysis:** Motivated by this finding, **we explore the potential benefits of incorporating the phase spectrum feature into blending-based detectors.** We hypothesize this would enhance performance in both cross-data and cross-manipulation evaluations.
        *  **Cross-data evaluation:** To validate our hypothesis, we first conduct an experiment in which we integrate the spectrum feature into the FWA detector, resulting in an improved FWA (iFWA). The experimental results, summarized in **Table 3 of the revised supplementary**, show a significant improvement achieved by iFWA (7.19% in Average AUC).
        * **Cross-manipulation evaluation:** Second, we conduct experiments and show the cross-manipulation outcomes achieved by iFWA in **Table 4 of the revised supplementary (5% improvement).** These visualizations serve to strengthen our argument about the consistent performance of iFWA.

We have added this analysis into **the updated supplementary, Line 462-479.** In the future, we will conduct more analyses and reveal more interesting insights based on our benchmark.

---

### Comment · Area_Chair_rLLu · 2023-08-29
**Please review the rebuttal**

Dear Reviewers and Ethics Reviewers,

The author-reviewer discussion period will end in one day. For those reviewers who have not yet review the authors' rebuttals, please do so as soon as possible and consider whether any rating adjustments are necessary. Thank you!

Thanks,
AC

---

### Decision · Program_Chairs · 2023-09-22

**Decision:**

Accept (Poster)

**Comment:**

Basically, most reviewers gave positive rating to this work. In the rebuttal, the issues raised by the reviewers have been well-addressed. Hence, I believe that this work can be accepted. The authors should modify the paper or possibly the project according to the comments, especially the ones common among multiple reviewers like the comparison with other codebases.